

# Hidden impacts of conservation management on fertility of the critically endangered kākāpō

Andrew Digby[1], Daryl Eason[1], Alejandro Catalina[2], Michael Lierz[3], Stephanie Galla[4,5], Lara Urban[6,7], Marissa F. Le Lec[6,8], Joseph Guhlin[6,8], Tammy E. Steeves[4,9], Peter K. Dearden[6,8], Tineke Joustra[10], Caroline Lees[11], Tane Davis[12], Deidre Vercoe[1] and Kākāpō Recovery Team

[1] Kākāpō Recovery Programme, Department of Conservation, Invercargill, New Zealand
[2] Department of Computer Science, Aalto University, Espoo, Finland
[3] Clinic for Birds, Reptiles, Amphibians and Fish, Justus-Liebig University Giessen, Giessen, Germany
[4] School of Biological Sciences, University of Canterbury, Christchurch, New Zealand
[5] Department of Biological Sciences, Boise State University, Boise, ID, United States of America
[6] Genomics Aotearoa, Dunedin, New Zealand
[7] Department of Anatomy, University of Otago, Dunedin, New Zealand
[8] Department of Biochemistry, University of Otago, Dunedin, New Zealand
[9] Genomics Aotearoa, Christchurch, New Zealand
[10] Unaffiliated, Auckland, New Zealand
[11] IUCN SSC Conservation Planning Specialist Group, Auckland, New Zealand
[12] Te Rūnanga o Ngāi Tahu, Christchurch, New Zealand

Corresponding author
Andrew Digby, adigby@doc.govt.nz

## ABSTRACT

**Background**. Animal conservation often requires intensive management actions to improve reproductive output, yet any adverse effects of these may not be immediately apparent, particularly in threatened species with small populations and long lifespans. Hand-rearing is an example of a conservation management strategy which, while boosting populations, can cause long-term demographic and behavioural problems. It is used in the recovery of the critically endangered kākāpō (*Strigops habroptilus*), a flightless parrot endemic to New Zealand, to improve the slow population growth that is due to infrequent breeding, low fertility and low hatching success.

**Methods**. We applied Bayesian mixed models to examine whether hand-rearing and other factors were associated with clutch fertility in kākāpō. We used projection predictive variable selection to compare the relative contributions to fertility from the parents' rearing environment, their age and previous copulation experience, the parental kinship, and the number of mates and copulations for each clutch. We also explored how the incidence of repeated copulations and multiple mates varied with kākāpō density.

**Results**. The rearing status of the clutch father and the number of mates and copulations of the clutch mother were the dominant factors in predicting fertility. Clutches were less likely to be fertile if the father was hand-reared compared to wild-reared, but there was no similar effect for mothers. Clutches produced by females copulating with different males were more likely to be fertile than those from repeated copulations with one male, which in turn had a higher probability of fertility than those from a single copulation. The likelihood of multiple copulations and mates increased with female:male adult sex

ratio, perhaps as a result of mate guarding by females. Parental kinship, copulation experience and age all had negligible associations with clutch fertility.

**Conclusions**. These results provide a rare assessment of factors affecting fertility in a wild threatened bird species, with implications for conservation management. The increased fertility due to multiple mates and copulations, combined with the evidence for mate guarding and previous results of kākāpō sperm morphology, suggests that an evolutionary mechanism exists to optimise fertility through sperm competition in kākāpō. The high frequency of clutches produced from single copulations in the contemporary population may therefore represent an unnatural state, perhaps due to too few females. This suggests that opportunity for sperm competition should be maximised by increasing population densities, optimising sex ratios, and using artificial insemination. The lower fertility of hand-reared males may result from behavioural defects due to lack of exposure to conspecifics at critical development stages, as seen in other taxa. This potential negative impact of hand-rearing must be balanced against the short-term benefits it provides.

# INTRODUCTION

## Factors affecting fertility in conservation-managed populations

Conservation strategies for wild-living threatened species rely on improving survival and productivity to increase population growth. Methods such as habitat restoration and predator control are used to enhance survival, but it is often problems with reproductive output which most limit recovery (*Bunin, Jamieson & Eason, 1997*; *Gage et al., 2006*; *Comizzoli & Holt, 2019*) and can have wide-ranging implications (*Findlay, Holland & Wong, 2019*). Management techniques used to address these problems include translocations, supplementary feeding and artificial insemination (*Lloyd & Powlesland, 1994*; *Castro et al., 2003*; *Houston et al., 2007*; *Armstrong & Seddon, 2008*; *Blanco et al., 2009*; *Heber et al., 2012*; *Dogliero et al., 2017*; *Schneider et al., 2019*). However, there has been little study of whether the conservation actions used to promote population growth of threatened species can in fact themselves impact productivity. This is at least partially due to any unintended consequences not being immediately apparent, especially in threatened species for which the ability to recognise significant trends is hampered by small sample sizes (*Garamszegi, 2016*). Here we consider factors which can affect fertility in conservation-dependent species, including the conservation management actions intended to improve population growth.

Hand-rearing, in which animals are raised in captivity by humans, is often used in threatened species conservation programmes (*Klusener et al., 2018*), primarily to increase productivity by improving survival during development to maturity (*Alagona, 2004*; *Heezik et al., 2005*). However, this intervention can have negative impacts, mainly by reducing long-term survival (*Aourir et al., 2013*; *Hampson & Schwitzer, 2016*; *Farquharson, Hogg &*

*Grueber, 2021*) and introducing behavioural issues (*Utt et al., 2008*; *Jones, 2008*; *Pacheco & Madden, 2021*) which may cause hand-raised individuals to be unsuited to life in the wild (*Meretsky et al., 2000*). These behavioural differences appear to affect productivity in some taxa (*King & Mellen, 1994*; *Beck & Power, 1988*; *Hampson & Schwitzer, 2016*), although the impacts are poorly understood in wild bird species (*Assersohn, Brekke & Hemmings, 2021*).

Mating behaviour, in terms of the number of mates and copulations, can directly affect fertility in birds. Females can increase the likelihood of egg fertilisation through polyandry—the 'fertility assurance hypothesis' (*Birkhead, Atkin & Møller, 1987*; *Reding, 2014*; *Rivers & DuVal, 2019*; *Santema, Teltscher & Kempenaers, 2020*)—and by copulating repeatedly with a single male (*Zhang et al., 2019*). These behaviours are influenced by adult sex ratio (*Grant & Grant, 2019*; *Birkhead & Montgomerie, 2020*): when competition is high, females in some species use repeated copulations to 'guard' preferred males and copulate with alternative males when their preferred choice is not available (*Petrie et al., 1992*).

Age affects reproductive output in some bird species (*Murgatroyd et al., 2018*; *Brown, Keefer & Songsasen, 2019*), but not others (*Zhang et al., 2014*; *Fay et al., 2020*) and in general is poorly studied in wild birds. Mating experience can also affect reproductive output: evidence suggests that both males and females with a greater number of previous breeding attempts may have higher reproductive success (*Kokko, 1997*; *DuVal, 2012*; *Assersohn, Brekke & Hemmings, 2021*), and so are preferred as mates (*Kokko et al., 1999*; *Jouventin, Lequette & Dobson, 1999*). Diet is also an important factor in avian reproductive output (*Selman & Houston, 1996*; *Klasing, 1998*), but this has also not been studied in most wild bird species (*Klasing, 1998*; *Assersohn, Brekke & Hemmings, 2021*). Fertilisation failure and very early embryo death can also result from increased homozygosity due to matings between closely-related individuals (*Hemmings, West & Birkhead, 2012*; *Assersohn et al., 2021*).

## Kākāpō

Low productivity limits population recovery of the kākāpō (*Strigops habroptilus*), a critically endangered, nocturnal and flightless parrot which is endemic to Aotearoa/New Zealand. Infrequent breeding, high infertility and low hatching success have hampered conservation efforts (*Clout, 2006*), although intensive management increased the population from 51 in 1995 to approximately 200 individuals in 2022. Remnant populations of kākāpō were translocated to predator-free island sanctuaries in the 1980s (*Powlesland et al., 1995*), and breeding has since occurred on five refuge sites: Whenua Hou/Codfish Island, Te Hauturu-o-Toi/Little Barrier Island, Te Hoiere/Maud Island, Pearl Island and Pukenui/Anchor Island (*Elliott et al., 2006*). All kākāpō are free-living in the wild, except during hospitalisation or rearing for some individuals.

Kākāpō breeding occurs irregularly, synchronised with the mass-fruiting (masting) of certain tree species, particularly the rimu tree (*Dacrydium cupressinum*). Rimu masts every 2–4 years (*Harper et al., 2006*) and is the predominant food fed to chicks when available (*Cottam, 2010*). The kākāpō is the only parrot species with a lek mating system (*Merton, Morris & Atkinson, 1984*): females visit leks to choose and copulate with

displaying males (*Eason & Moorhouse, 2006*), and both sexes often copulate with multiple partners. Females typically lay 2–3 eggs per clutch (range = 1–5). Males do not contribute to incubation or care of offspring. We refer to males as 'mates' if they have copulated with a female; it does not imply a pair association.

### Low fertility in kākāpō

A primary reason for low productivity in kākāpō is the high rate of infertility. Approximately 40% of kākāpō eggs are considered infertile from visual inspection ('candling'), although a recent fluorescence microscope study showed that 72% of these 'apparently infertile' kākāpō eggs were actually fertile, and instead failed due to very early embryo death (*Savage et al., 2021*).

There are a number of factors which may contribute to low fertility in kākāpō. With a small founding breeding population of 35 individuals and low levels of genetic diversity, inbreeding may be an important contributor (*Bergner et al., 2016*; *Dussex et al., 2018*; *Dussex et al., 2021*; *Guhlin et al., 2022*). Decreased female heterozygosity is correlated with lower hatching success and smaller clutch size in kākāpō (*White et al., 2014*), but male heterozygosity has no apparent effect on fertility, perhaps because males with the lowest heterozygosity may not mate at all (*White, 2012*).

Rearing environment may also influence fertility in kākāpō. All copulations occur in the wild, but eggs are often incubated artificially to maximise hatching success, and chicks are removed for hand-rearing if their life is at risk. This hand-rearing has caused behavioural issues, with two male chicks reared individually in 1997 and 1998 displaying imprinting on humans (*Eason & Moorhouse, 2006*).

Repeated copulations and multiple mates could affect fertility in kākāpō, as it does in other species (*Török et al., 2003*; *Santema, Teltscher & Kempenaers, 2020*). Repeated copulation in lekking species can also provide a strong test of theories for polyandry (*Parker & Birkhead, 2012*; *Rivers & DuVal, 2019*).

As a long-lived species with a life expectancy of several decades, kākāpō might experience age-related changes in reproductive output. Young age is a barrier to fertility: both sexes can mate from five years old, but no males younger than eight have produced fertile clutches. Impacts towards the end of life are less clear, since the age of kākāpō discovered as adults cannot be determined (*Horn et al., 2011*), but *White (2012)* found no impact of male age on egg fertility.

As a lek-breeding species, there is a high skew in kākāpō reproductive success, with a small number of males dominating copulations (*Eason et al., 2006*). The subsequent large variation in mating experience may also affect fertility.

Kākāpō are provided with supplementary food during breeding years to optimise productivity (*Elliott, Merton & Jansen, 2001*; *Clout, Elliott & Robertson, 2002*) and improve chick survival. Feeding increases clutch size and the proportion of females nesting and leads to a higher likelihood of mothers successfully rearing chicks, but there is no evidence that it affects fertility (*Elliott, Merton & Jansen, 2001*; *Houston et al., 2007*). Diet is not considered in this study because supplementary food contributes a low proportion of daily

metabolised energy (*Bryant & Bryant, 2006*), and incomplete feeding records and sharing of food stations makes it difficult to determine individual consumption over many years.

Other factors which can affect productivity in birds include injury, disease, stress, hormonal disruption, pollution and climate change (*Assersohn, Brekke & Hemmings, 2021*; *Assersohn et al., 2021*). These were not included in the current study since they were not considered important in wild kākāpō living on remote islands, and because the diseases which affect kākāpō do not appear to impact reproduction (*Gartrell et al., 2005*; *Jakob-Hoff, Potter & Shaw, 2009*; *Jakob-Hoff & Gartrell, 2010*).

Despite low fertility being one of the primary reasons for slow growth in the kākāpō population (*Elliott et al., 2006*), few studies have investigated its causes, and none have been multi-factorial. This study presents the first assessment of the relative impacts of multiple factors on kākāpō fertility, including life history, genetic and behavioural components. Our investigation focuses solely on fertility rather than other measures of productivity such as fledging rates because kākāpō eggs and chicks are subject to intensive management.

# MATERIALS AND METHODS

## Kākāpō management
### Copulation and nesting detection
Kākāpō are intensively monitored in order to maximise survival and productivity, with nearly every kākāpō fitted with a VHF radio transmitter since 1995. Initially these transmitters only allowed determination of location, so breeding behaviour was assessed manually. Copulation was detected by checking for sign at lek sites (feathers shed by the female during copulation), and nesting was inferred by daily triangulation (if adult females were repeatedly in the same location they were assumed to be incubating). Remote sensing methods improved the efficiency of collecting copulation data and their quality. Proximity sensors were installed at lek sites from 1997 to record male and female presence, and from 2012 the transmitters were fitted with activity sensors to provide copulation and nesting behaviour. The activity data were initially transmitted *via* coded VHF pulses to telemetry receivers used by field observers or mounted in an aircraft. Then from 2016, the activity data on the main breeding islands of Whenua Hou and Anchor Island were transmitted *via* a radio frequency data network connected to the island base and internet.

The use of VHF transmitters ensured that all nesting attempts since 1994 were detected, except for a very small number of cases when a female's transmitter failed. The addition of activity sensors in 2014 ensured that nearly all subsequent copulations were recorded. A small number of copulations were not detected by the transmitters due to hardware failure or unusual copulation activity, but subsequent nesting was detected. In addition, paternity of all offspring since 1997 was determined, first from microsatellite genetic testing (*Robertson et al., 2000*) and later from genotyping-by-sequencing of blood samples taken from fertile eggs or chicks.

Artificial insemination has been attempted in kākāpō during every breeding season since 2008, primarily to override genetically-unsuitable copulations. This is subject to significant logistical challenges, but in 2009 three chicks were produced by artificial insemination in

two clutches —a first for a free-living wild bird species. Subsequent attempts failed, until three successful inseminations produced three chicks in 2019, of which one fledged (KRT, 2021, pers. obs.).

### Fertility assessment

Fertility of eggs was assessed by trained observers using 'candling': a hand-held torch was used to illuminate the egg and inspect for signs of development (*e.g.*, embryo or blood vessels visible). This was conducted either in the nest or in an incubation facility, and was sufficient for detecting development from approximately four days after laying. Microscopic methods can detect earlier development (*Savage et al., 2021*), but these have only been conducted for a single breeding season for kākāpō, and so could not be used in the current study which spans multiple years. As a result of using 'apparent' rather than true fertility in our analyses, approximately a quarter of the eggs in which embryos died at a very young age (before four days) will have instead been classed as infertile (*Savage et al., 2021*).

### Nest management

From 1997–2019, most eggs (73%) were removed for artificial incubation, to increase hatching success, and replaced with 'dummy' eggs. A day or two before or after hatching, the eggs or chicks were returned to nests where possible, and closely monitored. Chicks were frequently cross-fostered among nests to increase the number and growth of chicks in nests. As as result, each chick may have had multiple foster mothers and often was not raised by its biological mother. Chicks fledged from nests at a mean of 73 days, but were still checked regularly until they were independent at around 219 days (*Farrimond, Clout & Elliott, 2006*).

### Hand-rearing

Artificial hand-rearing of kākāpō chicks was required due to health issues or if there were insufficient numbers of nests available (*Eason & Moorhouse, 2006*). In years when there was scarce natural food due to the rimu fruit not ripening, each nesting female could usually support only one chick, and surplus chicks were hand-reared. Between 1981 and 2019, 52% of chicks hatched were hand-reared for at least 10 days. To avoid imprinting on humans, chicks were not reared individually where feasible, and were usually kept in groups of 2–6 (*Eason & Moorhouse, 2006*). Where possible, chicks were reared on islands and then returned to nests, but some chicks required longer periods of hand-rearing. This long-term hand-rearing took place at a mainland facility, before the chicks were returned to islands at an age of approximately 80 days. Here they were weaned in large outdoor pens before being released into the wild at an approximate age of 120 days. Following fledging from the nest or from hand-rearing, most chicks were supported by supplementary feeding.

## Data collation
### Clutch data

Clutch data were collated from the Kākāpō Recovery Programme database for the breeding years between 1981 and 2019 (Table 1 and Data S1). The database contains all observed

**Table 1 Breeding attempts since modern records began in 1981.** Only data after 1990 were used in this study because data between 1981 and 1990 were incomplete. Note that for some clutches no copulations were recorded, and that fertility reported here is apparent fertility determined from candling; not true fertility from microscopic analysis. This is the full data set; some of these clutches were excluded from the fertility model. See text for further details.

| Year | Island | Clutches | Recorded copulations | Fertile eggs | Infertile eggs | Hatched | Fledged |
|------|--------|----------|---------------------|--------------|----------------|---------|---------|
| 1981 | Rakiura | 2 | 0 | 4 | 0 | 4 | 3 |
| 1985 | Rakiura | 3 | 0 | 3 | 6 | 2 | 0 |
| 1990 | Hauturu | 2 | 2 | 2 | 1 | 2 | 0 |
| 1991 | Hauturu | 4 | 3 | 6 | 2 | 4 | 2 |
| 1992 | Whenua Hou | 4 | 1 | 9 | 2 | 6 | 1 |
| 1993 | Hauturu | 2 | 3 | 1 | 3 | 1 | 0 |
| 1995 | Hauturu | 2 | 2 | 0 | 5 | 0 | 0 |
| 1997 | Whenua Hou | 6 | 6 | 7 | 5 | 4 | 3 |
| 1998 | Maud | 1 | 1 | 3 | 0 | 3 | 3 |
| 1999 | Pearl | 8 | 8 | 11 | 5 | 8 | 6 |
| 2002 | Whenua Hou | 24 | 34 | 42 | 25 | 26 | 24 |
| 2005 | Whenua Hou | 10 | 16 | 11 | 15 | 6 | 4 |
| 2008 | Whenua Hou | 5 | 12 | 10 | 0 | 8 | 6 |
| 2009 | Whenua Hou | 28 | 52 | 54 | 18 | 36 | 33 |
| 2011 | Anchor | 1 | 0 | 2 | 0 | 0 | 0 |
| 2011 | Whenua Hou | 8 | 13 | 14 | 4 | 11 | 11 |
| 2014 | Hauturu | 1 | 3 | 3 | 0 | 2 | 2 |
| 2014 | Whenua Hou | 7 | 14 | 6 | 9 | 5 | 4 |
| 2016 | Anchor | 22 | 32 | 32 | 38 | 21 | 15 |
| 2016 | Hauturu | 2 | 4 | 1 | 2 | 0 | 0 |
| 2016 | Whenua Hou | 20 | 31 | 30 | 19 | 26 | 20 |
| 2019 | Anchor | 37 | 60 | 56 | 67 | 42 | 37 |
| 2019 | Whenua Hou | 43 | 64 | 63 | 66 | 44 | 36 |
| Total | | 242 | 361 | 370 | 292 | 261 | 210 |

events for each individual, including transmitter activity data, captures, health checks, feeding records and copulations. These were combined with a dataset for each clutch since management began in 1981, containing clutch size, number of fertile eggs (apparent fertility), number of eggs hatched, and the number of chicks fledged, as well as paternity assumed from transmitter data and confirmed by genetic testing. Data prior to 1990 were excluded from the analysis since there was insufficient information for each nesting attempt. This yielded an initial data set of 237 clutches.

This data set contained first ($n = 197$), second ($n = 39$) and third ($n = 1$) clutches. Kākāpō will naturally re-nest if a nest fails early enough, and double clutching is used as a management method to improve productivity.

### Paternity assignment

Confirmation of paternity from genetic testing was available for 120 out of all 237 clutches laid from 1990–2019. Of the 117 clutches which did not have confirmed genetic paternity, it was necessary to identify the male which 'fathered' the clutch, so that its hand-rearing

status, age, copulation experience and the parental kinship could be compared to clutch fertility. Four clutches were excluded for which an unknown number of males copulated with the female, leaving 113 clutches without confirmed genetic paternity and 233 in total. In 92 clutches without genetic paternity confirmation, only one male copulated with the female, so assigning the 'father' was straightforward.

For a further 21 clutches (14 infertile, seven fertile) with unconfirmed genetic paternity, different identified males were confirmed or assumed to have copulated with the female. These clutches could not be excluded since doing so would remove the entire set (14) of infertile clutches produced by copulations with different males, biasing the clutches from multiple males to higher fertility by only leaving the fertile clutches. So to retain these 21 clutches, a 'father' was assigned from the 2–3 males identified to have copulated with the female, based on a likelihood of paternity from male copulation order. This likelihood of paternity was determined from clutches with confirmed genetic paternity, calculated as the proportion of clutches fathered by a male copulating first, last, middle, or first and last out of all the males which copulated with the female (Table S1). These probabilities were then used to select a 'father' from the candidate males using weighted sampling.

This method of selecting a clutch 'father' will have introduced errors due to the incorrect male being chosen in some cases, but these instances would have been few compared to the overall number of clutches. Moreover, this method would have caused less bias to the measured impact on fertility of copulations with multiple males compared to omitting the 21 clutches without genetic paternity confirmation. Furthermore, reducing the sample of clutches from multiple males would have greatly reduced the ability to assess the effect of sperm competition on fertility, which may be greater than the influence of the characteristics of the male which fathered the clutch. We acknowledge that the term 'father' cannot strictly be applied to an infertile clutch, but use it to signify the copulating male which had the highest likelihood of fertilising the eggs—and noting that in many cases, the eggs of these apparently infertile clutches were in fact fertilised.

A further seven clutches with mixed paternity and/or produced by artificial insemination were excluded, because these were not the product of a single male and female. The resulting 226 clutches were therefore the product of a single identified female and a male designated as the clutch 'father'. A further nesting attempt without any eggs was also removed, leaving 225 clutches from a total of 60 females and 51 males.

### Rearing status

For each clutch, the hand-rearing history of the mother and father was established from database records. Kākāpō were assigned as hand-reared if they had spent more than 10 days being hand-reared, at any period of their development; otherwise they were classed as wild-reared. A binary hand-rearing variable was chosen over a continuous one as it is more practicable to apply to management and because it simplified the statistical analysis. The binary variable was also more suited to the bimodal distribution of hand-rearing periods, with kākāpō chicks tending to be hand-reared for either a short period or for most of their development (Fig. S1). Many chicks are hand-reared for just a few days to enable them to recover from ill health or weight loss, particularly between the ages of two to three

weeks, when chicks fed by mothers receiving supplementary food often require removal to hand-rearing for a change of diet for up to five days. Alternatively, prolonged ill-health or lack of available nests means that they are hand-reared until they reach weaning age. Of the 111 adult kākāpō which contributed to the 225 clutches, 59 were hatched after intensive management and hand-rearing began. Of these, 21 (36%) were hand-reared for up to 10 days and 38 (63%) for more than 10 days, with only four hand-reared for between 10 and 60 days (Fig. S1). Hand-rearing could start at any chick age, so the number of days hand-reared was not necessarily the same as the age of the chick.

### Age assignment

The ages of the male and the female producing the clutch were calculated from hatch dates if these were known. Kākāpō of unknown age comprised 17 of the 60 females and 22 of the 51 males which contributed to the 225 clutches. These were assigned a minimum age of 10 years at discovery, which is a typical age of first breeding for males and females. Although the inclusion of the kākāpō of unknown age will have introduced errors due to inaccuracies in these estimated ages, these were likely to have been relatively small compared to the absolute ages at breeding, and the alternative of omitting these individuals would have resulted in greater model uncertainty due to the smaller sample size. This age assumption results in the oldest kākāpō breeding at 48.5 years of age (Fig. S2), which is younger than the presumed mean life expectancy in the contemporary managed population. However, the remnant populations from which the kākāpō of unknown age were sourced were under extreme predation pressure (*Karl & Best, 1982*; *Atkinson & Merton, 2006*), so would have had shorter life expectancies than the current protected population.

### Copulation experience

The previous copulation experience for each kākāpō was obtained from recorded copulation attempts and genetic paternity analysis. This provided an estimated cumulative number of copulations for the clutch mother and father prior to the clutch, summed over the lifetime of each individual, or since recordings began. The paternity analysis gave evidence of at least one copulation in cases when none were recorded. This estimated number of copulations was a lower limit, since not all copulations were detected—even with the electronic mating detection system—and since it was assumed that all founder individuals had not previously copulated at the time of their discovery. This underestimate was unavoidable given the lack of observational data prior to their discovery.

### Parental kinship

Pairwise kinship for all male–female combinations of living and recently-deceased kākāpō were obtained from a pedigree generated from the kākāpō studbook in PMx (*Lacy, Ballou & Pollak, 2012*). To address the assumption of founders being equally unrelated to one another (*Ballou, 1983*), founder relatedness was incorporated into the kākāpō studbook using genomic-based estimates of relatedness. In this process whole genome resequencing data from 169 birds was used to discover SNPs using the reference-guided Deep-Variant pipeline (*Poplin et al., 2018*). A stringent filtering protocol using BCFTools (*Li et al., 2009*) and VCFTools (*Danecek et al., 2011*) was applied to include biallelic SNPs with a
minimum coverage of three, a maximum coverage of 100, a minimum Phred quality score of 10, a genotyping rate > 90%, a minor allele frequency of 0.05, and pruning for linkage disequilibrium with an $r^2$ of 0.8 and a sliding window of 1000 sites. This filtering resulted in 8,407 high confidence markers with high depth (average = 19.88 ±8.08 SD) and low missing data (average = 0.0002 ±0.0001 SD) across individuals. Initial testing was performed to evaluate estimators for accuracy and precision with mother-offspring relatedness, including: KING (*Waples, Albrechtsen & Moltke, 2019*, estimated through the package NGSrelateV2, *Hanghøj et al., 2019*), KGD (*Dodds et al., 2015*), KGD with a correction for self-relatedness (as per *Galla et al. 2020*), Rxy (*Hedrick & Lacy 2015*, estimated through NGSrelateV2), and TrioML (*Wang 2007*, estimated through the R program `related`, *Pew et al. 2015*). Rxy was chosen as the best relatedness estimator, given its high accuracy for mother-offspring relatedness and the benefit of bounding between 0–1 for ease of entry into PMx (*Lacy, Ballou & Pollak, 2012*). Final relatedness estimators were calculated between the 35 founders identified in the kākāpō studbook and were incorporated into PMx as kinship (half of the relatedness value). Parental kinship for the clutches in this study were produced in PMx using the founder-corrected studbook. These values were in the range 0–0.265, with a median of 0.0074 and a distribution that was positively skewed (Fig. S2).

## Statistical analyses
### Bayesian model structure
A Bayesian generalised linear mixed model was used to assess factors contributing to clutch fertility. The explanatory variables considered to have potential effects on clutch fertility were chosen from data exploration and knowledge of kākāpō ecology. These were: the age, hand-reared status and previous copulation experience (number of previous copulations) of both clutch mother and father; the copulation behaviour of the clutch mother, in terms of the number of copulations and the number of different males the female copulated with to produce the clutch; and the parental kinship.

The hand-rearing status of the clutch mother and father was set to a binary variable: one if the individual had been hand-reared for more than 10 days and zero otherwise. The female copulation behaviour was a categorical variable with three levels: one copulation with one male, more than one copulation with the same male, and copulations with different males. This latter category contained clutches in which a female copulated more than once with at least one of the multiple males ($n = 17$). Parental kinship was a continuous variable in the range 0 –0.265. Copulation experience was defined as the number of previous copulations detected prior to those which yielded the clutch, since records began. This was calculated for both the female and male which produced the clutch.

The numeric explanatory variables were scaled and centred to have mean of one and standard deviation of 0.5 (*Gelman et al., 2008*); the categorical variables were defined as factors. No interactions of the covariates were considered relevant. Collinearity of predictors was examined with correlation plots and paired posterior plots: no significant correlation among predictors were found, so none were excluded.

**Table 2 Model predictors.** Parameters for the 217 clutches used in the Bayesian model relating clutch fertility to the characteristics of the clutch mother and clutch father (the male and female which produced the clutch). See Fig. S2 for distributions of the numeric variables.

| Component | Variable | Type | Values (frequency) |
|---|---|---|---|
| Response | Clutch fertility | Binary | 0 (80) / 1 (137) |
| | Mother hand-reared | Logical | true (64 clutches; 26 females) / false (153 clutches; 34 females) |
| | Father hand-reared | Logical | true (43 clutches; 12 males)/false (174 clutches; 38 males) |
| | Mother age (years) | Continuous | range = 4.8–48.5, mean = 20.9, median = 17.8 |
| | Father age (years) | Continuous | range = 4.8–43.4, mean = 22.7, median = 20.8 |
| Fixed | Mother copulation behaviour | Categorical | 1 copulation (104)/1 male, > 1 copulation (50)/Different males (63) |
| | Mother previous copulations | Integer | range = 0–17, mean = 4.4, median = 4 |
| | Father previous copulations | Integer | range = 0–33, mean = 6.5, median = 4 |
| | Mother/father kinship | Continuous | range = 0 –0.265, mean = 0.021, med = 0.0074 |
| | Mother | Categorical | 60 individuals, 1 –9 repeats, mean = 3.6, median = 3 |
| Random | Father | Categorical | 50 individuals, 1 –16 repeats, mean = 4.3, median = 4 |
| | Year | Categorical | range = 1990 –2019, 16 levels |

The response variable was the binary fertility status of each clutch (0/1), with a Bernoulli error distribution. This was used instead of the proportion of eggs in a clutch that were fertile, because the fertility of each egg was not independent of the fertility status of others in the clutch (Fisher exact test for association between categorical variables, $p < 0.001$, odds ratio = 0.0153, [0.00833, 0.0270] 95% confidence interval). Of 602 eggs in clutches with more than one egg, 313/332 fertile eggs were in a clutch with other fertile eggs, and 216/270 infertile eggs were in infertile clutches (Data S1).

Random effects were included for clutch mother and father identity to account for pseudo-replication, and for year, to account for unmeasured environmental variation. No effect was included for island, since this predictor was highly imbalanced, with two of the five breeding islands dominating the number of clutches: Whenua Hou (145) and Anchor Island (59) produced 91% of the 225 clutches.

Observations with missing values for any of the predictors were excluded. From the initial set of 225 clutches, the final model data contained 217 clutches with complete values for all eleven input variables (Table 2). This resulted in a mean of 19.7 events per variable, which was greater than the minimum of 10–15 recommended for linear regression modelling (*Heinze, Wallisch & Dunkler, 2018*).

### Bayesian model variable selection

Small datasets are common in threatened species research, leading to statistical challenges such as low precision, low accuracy and instability masking true relationships between variables (*Garamszegi, 2016*). To prevent the model from overfitting to the data due to the large ratio between number of parameters and number of observations, it is often necessary to limit the number of variables in the model (*Heinze, Wallisch & Dunkler, 2018*). Methods such as penalized regression and shrinkage priors are commonly used to this effect (*Piironen & Vehtari, 2017b*; *Vehtari, Gelman & Gabry, 2017*; *Erp, Oberski*

& *Mulder, 2019*; *Carvalho, Polson & Scott, 2010*; *Hastie, Tibshirani & Wainwright, 2015*; *Narisetty & He, 2014*). However, these methods do not really produce truly sparse solutions, as every variable has a non-zero probability of inclusion. Instead, we applied projection predictive variable selection (*Piironen, Paasiniemi & Vehtari, 2020*; *Catalina, Bürkner & Vehtari, 2020*), which effectively selects a subset of variables from a previously fitted reference model. This method ranks the variables in order of their contribution to the model predictions, replacing the posterior of the model with a constrained projection which provides predictive performance equivalent to the full model (*Piironen, Paasiniemi & Vehtari, 2020*; *Catalina, Bürkner & Vehtari, 2020*), as measured by the Kullback–Leibler divergence of their predictions (*Goutis, 1998*). Projection predictive variable selection has been shown to outperform other more established variable selection methods (*Piironen & Vehtari, 2017a*). Furthermore, it can be applied not only to generalised linear models, but also to generalised linear and additive multilevel models, allowing the projection of random (additive) effects.

In order to rank the variables during model search, projection predictive variable selection uses forward search for multilevel or additive models and a faster L1-like heuristic for generalised linear models. Since the model structure included random effects per individual in the sample, we restricted the search to first select the fixed effects, and only then added the random effects. This was to ensure that the predictive variance would not be completely saturated by the individual random effects and properly measure the effect of the biologically-relevant terms.

### Bayesian model execution and validation

All analyses were conducted in R (version 4.1.2; *R Core Team 2020*), with the Bayesian model implemented in R package brms version 2.16.3 (*Bürkner, 2017*) and projection predictive variable selection applied with package projpred version 2.0.5.9 (*Piironen, Paasiniemi & Vehtari, 2020*). A regularised horseshoe prior was used (*Piironen & Vehtari, 2017b*), with one degree of freedom for the student-T prior for the local and global shrinkage parameters, and a scale of one for the global shrinkage and regularisation parameter (*Bürkner, 2017*). The model was run with four chains, with 15,000 iterations and 15,000 warm-up iterations per chain. Model code and results are available in Data S1.

Projection predictive variable selection was then used to provide a reduced model with equivalent predictive performance to the full model. The variables included in the reduced model were selected by the improvement they provided to the model. As criteria for the selection of variables we checked the ELPD improvement and each variable's marginal posterior, and selected those whose posterior mass was clearly non-zero and whose ELPD improvement was significant.

Model validity was assessed by Pareto $k$ estimates (*Vehtari, Gelman & Gabry, 2017*; *Vehtari et al., 2019*), and by graphical residual and posterior predictive checks using the bayestestR package (*Makowski, Ben-Shachar & Lüdecke, 2019*). The relative influence of each predictor on clutch fertility was assessed by Bayesian indices of effect existence and significance (*Makowski et al., 2019*). Effect existence was measured by the probability of direction (p.d.), which is the proportion of the posterior that is of the same sign as the

median and is interpreted as the probability that a variable is positive or negative (*Makowski et al., 2019*). The effect 'significance' was assessed from the amount of intersection of the full posterior distribution of the constrained projection with the region of practical equivalence (ROPE; *Makowski et al. 2019*). This region of 'practically no effect' provides a measure of the 'importance' of a parameter, based on the proportion of the posterior which overlaps the ROPE. It is quantified by the probability of significance (p.s.): the proportion of the distribution outside the ROPE. If there were values of the distribution both above and below the ROPE, the probability of significance was reported as the higher probability of a value being outside the ROPE. A range of $[-0.18, 0.18]$ was used for the ROPE, as recommended for logistic models (*Kruschke & Liddell, 2018*).

### Multiple copulations and population density

In addition to the Bayesian fertility model, we also investigated the incidence of multiple copulations with kākāpō abundance. We merged repeated copulations with one male and copulations with different males into a single category of 'multiple copulations', in order to achieve sufficient sample sizes. We correlated the proportion of clutches produced by multiple copulations with the total number of adult female and male kākāpō, and the adult sex ratio, on Whenua Hou for each year since 1990. This analysis was confined to a single island to avoid inter-island effects, and Whenua Hou was chosen as it produced a large proportion of all clutches from 1990–2019 (64%). Due to the size of the island and the lek breeding system of kākāpō, there is opportunity for copulation between all breeding-aged males and females. Correlations were assessed using the `correlation` package (*Makowski et al., 2020*) in R, using the Pearson correlation coefficient and Holm adjustment method (*Holm, 1979*).

## RESULTS

### Factors affecting fertility

Projection predictive variable selection in the Bayesian mixed model showed that of the fixed terms, the hand-rearing status of the clutch father explained most of the variance in the model, followed by the copulation behaviour of the mother (Fig. 1, Data S2). These two fixed terms made the biggest change in expected log predictive density (ELPD) difference, contributing 15% and 11% respectively of the total difference in ELPD; all other fixed terms contributed just 5% combined. These proportions should only be used as a guide to the relative contribution to the model variance, since the ELPD depends on the order of the projected terms. Clutch father hand-rearing status and mother copulation behaviour were the only two fixed terms which had projected posterior distributions distinguishable from zero (Fig. 2). All other fixed terms had negligible impact on the model fit, and had projected posterior distributions indistinguishable from zero (Figs. 1 and 2). Of the random terms, clutch father and mother identity contributed most to the variance (50% and 12% of the total ELPD variation, compared to 8% for the year random term), with father identity the most important of all fixed and random parameters. Random effects dominating fixed effects is common in mixed models, and can obscure the underlying fixed model structure. It suggests that there was substantial variation in the model due to individual effects which

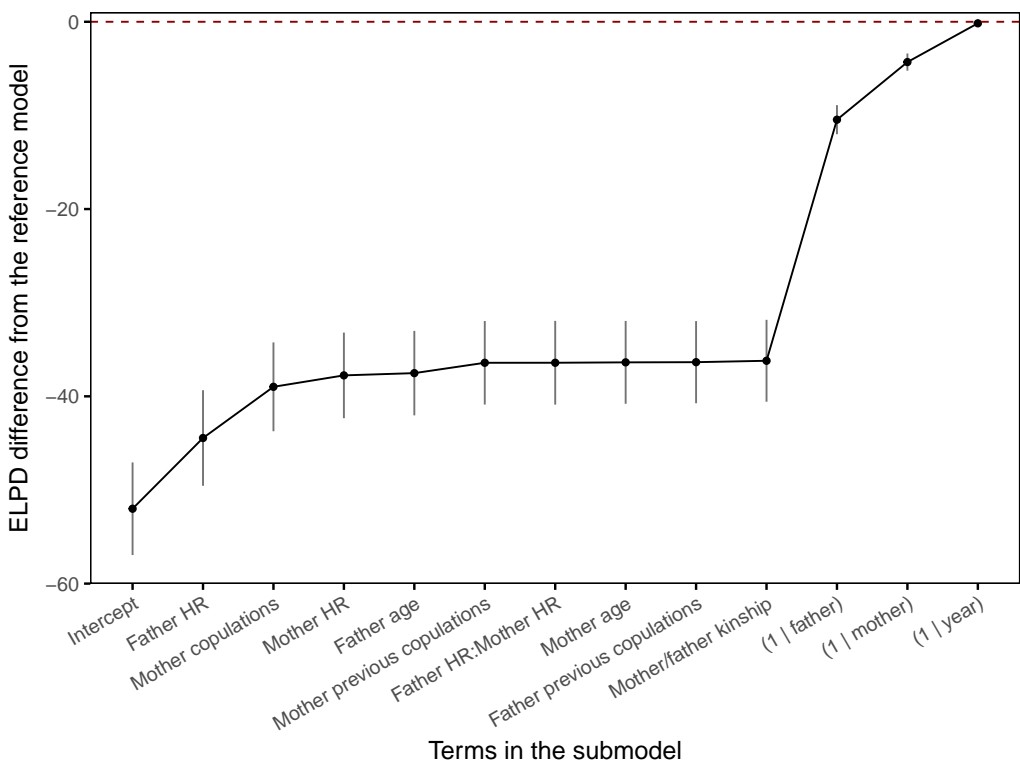

**Figure 1  Projection predictive variable selection results.** Variables ranked by their contribution to the fertility model's predictive ability, measured by the change each makes to the expected log predictive density (ELPD). As each variable is added from left, the change in ELPD difference from the previous term shows the change in the model's performance, relative to the full model. Fixed terms are ordered in their contribution to the model variance, with random terms selected last. The dashed line shows the ELPD for the full model. The reduced model containing the fixed variables of clutch father rearing status and mother copulation behaviour ("Mother copulations"), with random terms for clutch father, mother and year, provided equivalent predictive performance to the full model. HR = hand-rearing.

were not captured by the fixed variables. A reduced model containing mother copulation behaviour, clutch father hand-rearing status and random terms for clutch father, mother and year provided predictive performance equivalent to the full model (Fig. 3). This reduced projected model explained approximately 31% (estimated $R^2 = 0.031$) of the total observed variation in clutch fertility.

Clutches from hand-reared fathers were associated with the highest change in clutch fertility, with a strongly significant negative effect (probability of direction, p.d. = 0.98, probability of significance, p.s. = 0.93 in the reduced model). The effect of females copulating with different males had similarly high importance, associated with a strongly positive and significant increase in fertility compared to single copulations (p.d. = 0.97; p.s. = 0.92). Clutches in which females copulated repeatedly with the same male were also highly likely to be more fertile than single copulations, but with lower significance (p.d. = 0.81, p.s. = 0.60; Fig. 3). The remaining fixed terms of clutch mother rearing status, clutch mother and father age, parental kinship, and clutch mother and father copulation

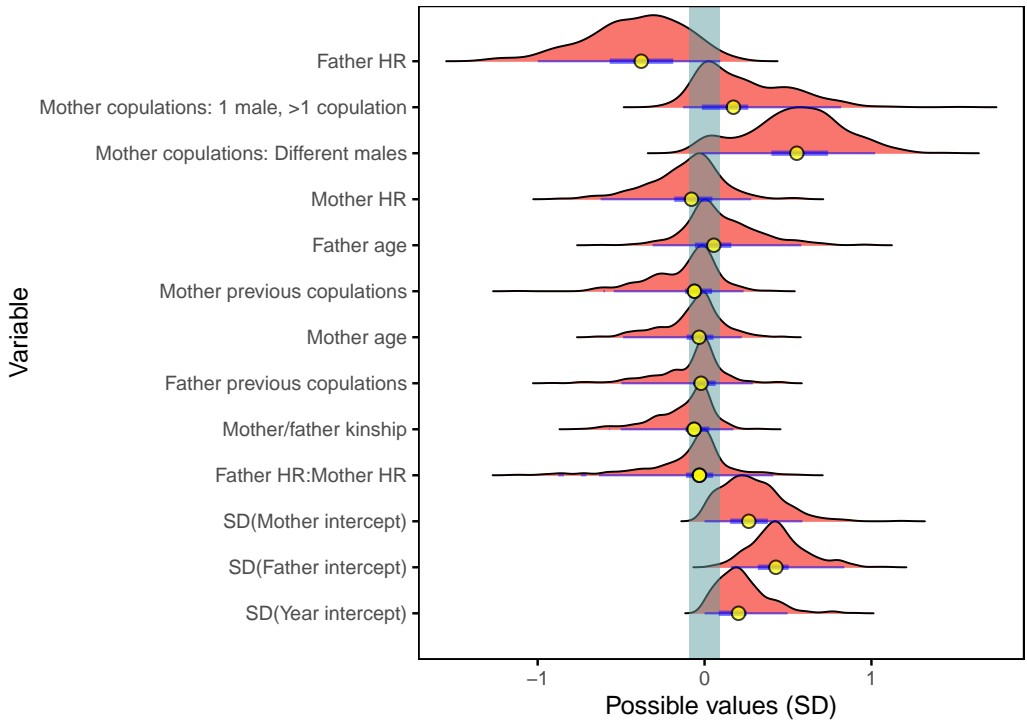

**Figure 2** **Posterior coefficient distributions of the coefficients for the full projected model.** The effect of each variable on predicted clutch fertility in the full model. The less a posterior distribution intersects the ROPE (region of practical equivalence, denoted by the shaded vertical bar), the stronger the association of that variable with fertility (see Statistical Analyses for details.) Distributions to the right of the ROPE indicate a positive impact on clutch fertility, and those to the left a negative impact. Posterior medians are shown by filled yellow circles, with thick and thin horizontal blue bars denoting the 50th and 95th percentiles respectively. Of the fixed effects, only the clutch mother copulation behaviour and clutch father hand-rearing variables had posteriors likely to be non-zero. The mother copulation behaviour variable is split into its factor levels, with the reference level a single copulation. For rearing status, wild-reared is the reference level. Considering the posteriors and the projection predictive variable selection results, only these two fixed variables were retained in the reduced model. HR = hand-rearing.

experience were not included in the reduced model as they all had a very low impact on clutch fertility compared to hand-rearing status of the clutch father and the copulation behaviour of the clutch mother.

Model predictions (Fig. 4) showed that females copulating with multiple males had a high probability of producing a fertile clutch, especially if the clutch father was wild-reared (84% for a wild-reared father and 66% for a hand-reared father). Females copulating repeatedly with a single male had a higher likelihood of clutch fertility than those copulating just once (72% *vs* 64% for a wild-reared mate, and 50% *vs* 39% for a hand-reared mate). Irrespective of the number of copulations and mates, copulating with a hand-reared male decreased the likelihood of clutch fertility compared to a wild-reared male.

## Multiple copulations and kākāpō density

The likelihood of females engaging in multiple copulations (either with the same male or different males) was strongly positively correlated (Pearson correlation, $r = 0.93$,
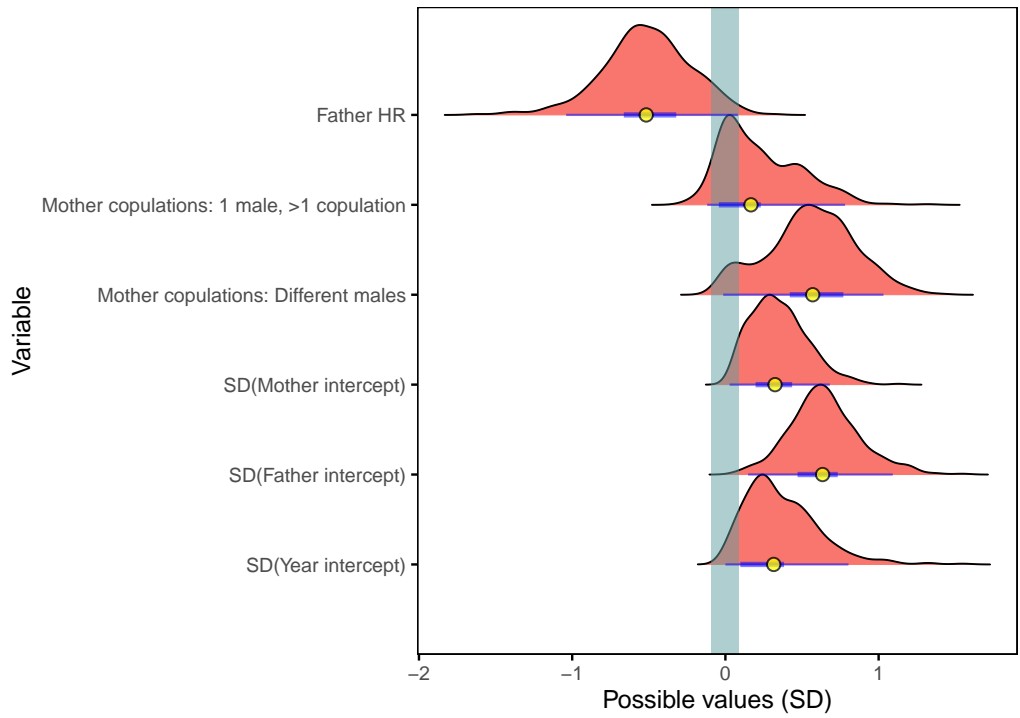

**Figure 3 Posterior distributions of the coefficients for the projected reduced model.** The effect on predicted clutch fertility of the subset of variables in the reduced model, which had predictive performance equivalent to the full model containing all terms. The reduced model contained all three random effects and the two fixed effects with the highest variance contribution: clutch father hand-rearing status and clutch mother copulation behaviour. Description and symbols as in Fig. 2.

95% CI [0.74–0.98], $p < 0.001$, $t = 7.44$, d.f. = 8) with the size of the adult female population on Whenua Hou from 1990–2019 (Fig. 5). The association between multiple copulations and male abundance was much weaker (Pearson correlation, $r = 0.61$, 95% CI [−0.02–0.90], $p = 0.059$, $t = 2.20$, d.f. = 8), but there was a strong correlation between multiple copulations and the female:male sex ratio (Pearson correlation, $r = 0.92$, 95% CI [0.71–0.98], $p < 0.001$, $t = 6.88$, d.f. = 8).

There was substantial variation in multiple copulation behaviour among females. Of the 60 females in the model data set, 38 (63%) copulated with different males in at least one breeding season, 27 (45%) had repeated copulations with the same male at least once, and 52 (87%) produced at least one clutch following a single copulation.

## DISCUSSION

Low hatching success, particularly due to egg infertility or very early embryo death, is one of the main obstacles to recovery for the critically endangered kākāpō. Using all available reproductive data for the species, this study shows that of those assessed, the dominant factors affecting clutch fertility are male hand-rearing status and female copulation behaviour, in terms of the number of copulations and number of mates.

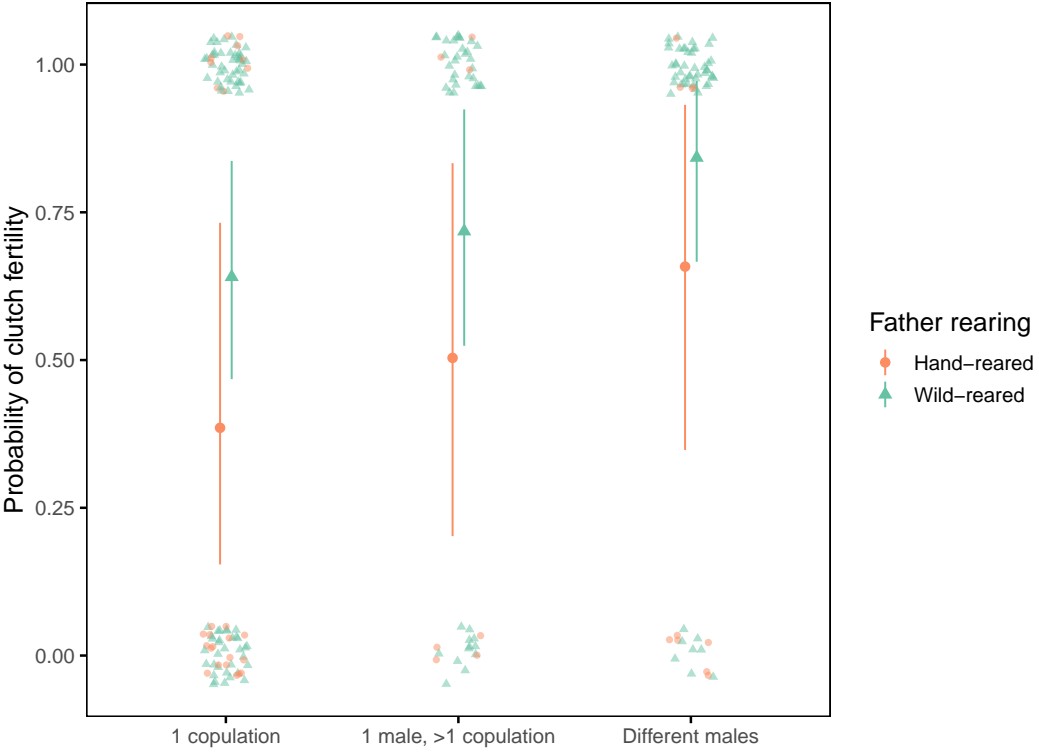

**Figure 4  Fertility model predictions for the interaction of clutch mother copulation behaviour and clutch father rearing status.** Predictions from the reduced model for how the likelihood of clutch fertility varied with the number of mates and copulations of the clutch mother, and with the rearing environment of the clutch father. A clutch is considered fertile if at least one egg is fertile and infertile if all eggs are infertile. Model predictions are shown as large filled circles, with 95% highest posterior density intervals denoted by vertical bars. Small filled symbols denote the observational data, with circles for hand-reared fathers, and triangles for wild-reared fathers. The data are jittered along both axes for clarity.

Fertility was reduced in clutches produced by a hand-reared father compared to a wild-reared father, increased if the mother copulated repeatedly with one male compared to a single copulation, and increased further still if the mother copulated with more than one male.

## Small samples sizes and longitudinal data

The sample size of 217 clutches in this study is statistically small, but represents a substantial and long-term monitoring effort utilising advanced technologies. Few wild species are monitored as intensively as the kākāpō, with individuals closely followed over decades and nearly all copulations recorded. This longitudinal data set has enabled analysis of potential impacts on fertility, highlighting the importance of adequate monitoring to assess effects of management methods which may not be immediately apparent, as well as the importance of long-term, individual-based studies (*Clutton-Brock & Sheldon, 2010*).

Despite this effort, the impact of small data sets must be considered when evaluating these results. Small sample sizes are often unavoidable in threatened species analyses, which can lead to imprecise, inaccurate or unstable results, and important effects being missed

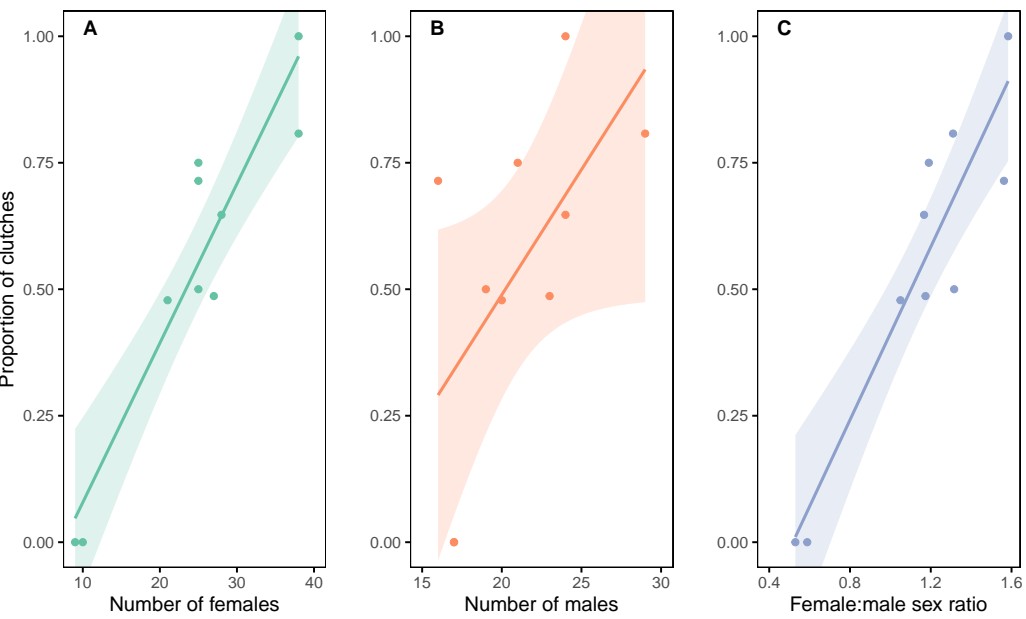

**Figure 5 Proportions of clutches with multiple copulations in relation to the number and sex ratio of adult kākāpō.** The association between the incidence of clutches produced by multiple copulations with (A) female and (B) male kākāpō abundance and (C) female:male sex ratio, on Whenua Hou over ten breeding seasons from 1990–2019. Clutches from multiple copulations are those produced by a female repeatedly copulating with a single male or copulating with multiple males. Lines and shading show linear regression fits with 95% confidence intervals. The scales for the number of kākāpō in panels (A) and (B) are different.

due to apparently non-significant results arising from high uncertainty (*Garamszegi, 2016*). This is why for small data sets it is important to use robust statistical methods which provide reliable uncertainty measures and can rank predictors by their contribution to the response, such as the Bayesian predictive projection variable selection utilised here. Even with these methods, the impact on fertility of the effects we report is likely to be underestimated. This must be considered when using these results to make conservation management decisions, and emphasises the importance of reanalysis when data sets become larger with further monitoring. A further benefit of the Bayesian methods employed here is that they make such reanalysis straightforward.

## Rearing environment

The model showed a strong impact of father hand-rearing status on clutch fertility, with a 98% probability that a hand-reared father had a negative effect on clutch fertility compared to a wild-reared father and a 93% probability that this effect was significant. This result provides a rare demonstration of hand-rearing affecting productivity in a bird species. In fact, evidence of similar effects across *any* taxon is extremely limited, in contrast to examples of the impact of captivity or rearing method on other traits such as survival (*Farquharson, Hogg & Grueber, 2018*). This is likely a result of the difficulty of measuring these effects, which usually requires longitudinal data of reproductive success across several generations

(*Clutton-Brock & Sheldon, 2010*), and which is compounded by a strong bias in fertility studies towards commercial bird species and a tendency to focus on male reproductive issues (*Assersohn, Brekke & Hemmings, 2021*).

### Limitations

We used a binary hand-rearing variable, but this does not mean that kākāpō hand-reared for fewer than the 10 day cut-off were immune from impacts of hand-rearing. Inclusion of these hand-reared individuals (albeit hand-raised for a very short period) may have reduced the influence of hand-rearing in the model, but we would expect this effect to be small, given that nearly all of these (20/21; Data S1) were hand-reared for five days or less. An individual was defined as having been hand-reared if it was hand-raised for at least 10 days at any stage during development. When the hand-rearing occurs may be as important as its duration, but this could not be assessed with the available data. Examples from other species demonstrate that the timing of imprinting varies among species, and that even a short hand-rearing period may influence behaviour (*Jones, 2008*). Male falcons reproduce less effectively if reared by hand for more than the first week of their life (*Lierz, 2019*), and in raptors imprinting or even partial imprinting can affect pair behaviour and therefore reduce egg fertility (*Jones, 2008*; *Lierz, 2008*). Whereas it is clear that a fully hand-raised bird might not be able to reproduce with conspecifics, there is uncertainty over the impact of shorter hand-rearing periods. It is feasible that any time during the development period that an individual is not raised by conspecifics might later lead to behavioural alterations (*Irwin & Price, 1999*). Assessing whether there is a particular period during development when the impact of hand-rearing is most pronounced should be a focus for future analyses when sufficient data are available.

### Implications

Examples from other taxa suggest that the reason for hand-rearing affecting clutch fertility in kākāpō is likely to be behavioural. In primates, lack of access to conspecifics lowers reproductive output through suspected behavioural mechanisms (*King & Mellen, 1994*; *Beck & Power, 1988*; *Hampson & Schwitzer, 2016*). We suggest that hand-reared male kākāpō have a lower ability to copulate successfully than their wild-reared counterparts as a result of sexual imprinting (*Irwin & Price, 1999*). Sexual imprinting on humans is known in other species such as falcons, with imprinted males showing no interest in mating with female birds (*Lierz, 2008*). There is qualitative evidence of this in kākāpō, with one individual hand-reared alone in 1997 (from three to 15 weeks of age) apparently unable to mate as a result of strong imprinting on humans (*Eason & Moorhouse, 2006*). Another male hatched in 1998 was also hand-reared alone for a similar period and is partially sexually imprinted on humans. Although this individual has mated with female kākāpō, it has not yet (to 2019) naturally produced fertile eggs. These imprinting behaviours appear to most strongly affect male chicks reared alone: females have been similarly hand-reared alone without any observed negative reproductive impacts, although these may be less immediately apparent (*Eason & Moorhouse, 2006*; *Harper & Joice, 2006*). The relative impact of hand-rearing birds individually rather than with conspecifics is demonstrated in other species. In falcons, for example, chicks hand-reared alone tend to be less successful

breeders and have more behavioural problems than those reared in a cohort (*Jones, 2008*). As a consequence, kākāpō chicks are no longer hand-reared separately from other individuals, unless it is unavoidable due to particular health issues, in which case the time that they are hand-reared without conspecifics is minimised (*Eason & Moorhouse, 2006*).

Hand-rearing has been shown to reduce reproductive output—although not fertility—in takahē (*Porphyrio hochstetteri*), a threatened rail endemic to New Zealand. Hand-raised takahē fledge approximately 50% fewer offspring than their wild-reared counterparts, even though egg fertility is similar (G Greaves, New Zealand Department of Conservation, 2015, pers. comm.). This suggests that hand-reared takahē have reduced chick-rearing ability and that a behavioural mechanism is responsible. While this does not directly support our hypothesis that hand-rearing affects male copulation ability in kākāpō, it does demonstrate that hand-rearing can strongly affect reproductive behaviour.

The evidence for negative impacts of hand-rearing on kākāpō reproductive output may have profound consequences for the conservation of the species. Hand-rearing is a key part of management, used to prevent loss of chicks which would naturally have died through starvation or ill health. More than half of the 261 chicks hatched from 1981–2019 were hand-raised for at least 10 days, usually in the first four weeks after hatching. Hand-rearing has made a stronger contribution to population growth than perhaps any other management method. There have been no other apparent negative effects of this practice: from 32 hand-reared females which bred up to 2019, 25 (78%) hatched chicks, and all of these fledged at least one chick.

Steps are already taken to avoid imprinting in kākāpō: chicks are not reared alone, are only hand-reared if there is no alternative and are released from captivity soon after weaning. But the additional impact on fertility identified here adds greater pressure to avoid hand-rearing of males. This is at odds with the current management policy which prioritises leaving female rather than male chicks in nests when there is insufficient capacity. This has been applied as it was assumed to be more important to produce high-quality, naturally-raised females, the availability of which was thought to be one of the primary factors limiting population growth.

## Female copulation behaviour

This study shows that female copulation behaviour—in terms of the number of copulations and mates—has a significant effect on clutch fertility in kākāpō. The mother copulation behaviour variable contributed more to the model variance and had a higher importance than any other fixed term except for the hand-rearing status of the clutch father. The model predictions showed a clear trend in the likelihood of clutch fertility with female copulation behaviour: lowest for clutches produced by a single copulation, higher for those from repeated copulations with one male, and highest for clutches produced by females copulating with multiple males. The effect of copulating with multiple males had a strongly positive and significant impact on clutch fertility (97% probability that it was positive and a 92% probability of significance). Furthermore, our results demonstrate that this female copulation behaviour was strongly influenced by the number and sex ratio of kākāpō in a

population. With more females and a higher female:male sex ratio, an increasing number of clutches were produced by multiple copulations, either with the same or different males.

### Limitations

Before assessing the implications of these findings, it is important to consider the limitations of the evidence. First, the estimated effect on fertility from copulation with multiple males would have been subject to errors from assigning a putative clutch 'father' to the 21 such clutches which had no genetic paternity confirmation. This would have most influence on the clutch father variables, because some infertile clutches may have been assigned the incorrect 'father' and therefore the incorrect hand-rearing status, age, copulation experience or parental kinship. However, omitting these clutches would instead have created a much larger impact on the multiple copulation effect by removing a greater proportion of infertile clutches (14/14 infertile clutches compared to 7/53 fertile clutches; Fig. 4), therefore overestimating the fertility increase from copulating with different males. Furthermore, this method made use of the available information of the identity of the 2–3 candidate fathers known to have copulated with the female (for example, in three infertile clutches all potential fathers had the same rearing status), which the alternative method of imputing missing values (*van Buuren & Groothuis-Oudshoorn, 2011*) would not. That this process affected less than 10% of all clutches also suggests that the impact of any incorrectly-assigned paternity was relatively small. This was confirmed by a comparison of model results with and without these clutches included, which showed that the overall conclusions were preserved.

There were other limitations due to the size and nature of the data. One is that the clutches from multiple mates included those with repeated copulations with at least one of the males, so that any effect attributed to multiple mates could at least partially be due to repeated copulations. The number of mates and number of copulations could be decoupled with a larger data set in future, and a continuous rather than categorical parameter used for the number of copulations. The use of a binary variable for clutch fertility, rather than the proportion of fertile eggs per clutch, similarly results in a loss of information, but is unavoidable given the non-independence of egg fertility within a clutch. The timings of copulations relative to egg laying and the stage of the breeding period were not considered in our analyses, but may be important predictors of fertility. It is also possible that the incidence of repeated copulations and multiple mates was a function of female condition, with those in better condition able to visit and copulate with more males. However, we did not include this effect since there were sparse data on female condition, and it is unlikely to have a strong impact because most breeding females were maintained within a narrow weight range by supplementary feeding (*Clout, Elliott & Robertson, 2002*).

The strong correlation of the proportion of clutches from multiple copulations with adult sex ratio could potentially be influenced by an unmeasured co-correlate, particularly one which has changed over time. The sex ratio has changed on Whenua Hou since 1990 (Data S3), largely due to an increase in the early 2000s which can be attributed to the optimisation of supplementary feeding resulting in more female chicks (*Clout, Elliott & Robertson, 2002*) However, we do not consider it feasible that this change in supplementary
feeding could have similarly affected female copulation behaviour. Temporal changes in spatial partitioning of the males and females on Whenua Hou is also unlikely to have contributed to the observed correlation, since females have always had access to all displaying males.

### Fertility assurance

The kākāpō reproductive data provide a rare opportunity to assess fertility benefits of females copulating repeatedly with the same male. Close observation of individual mating behaviour is rare in wild bird species, so there have been limited opportunities to assess the impact of repeated copulations to help determine the reason for this behaviour. Some of the hypotheses for repeated copulations require a pair bond or paternal investment, which are not present in kākāpō (*Hunter et al., 1993*). Other explanations are that repeated copulations could reduce the likelihood of the male copulating successfully with other females; could devalue the sperm from an inferior male; or could increase fertility through a higher likelihood of the female receiving sufficient sperm (*Petrie, 1992*; *Heeb, 2001*; *Hunter et al., 1993*). The first of these hypotheses is less likely to apply to kākāpō, because males copulate relatively infrequently, despite apparently having the capacity to do so more often (*Eason et al., 2006*). The second explanation is not supported by clutches in which the female kākāpō copulated only with one male (48% of the 217 clutches used in the model) or with one male before and after a second (9% of the 63 clutches with multiple mates). The final explanation, the increased fertility hypothesis (*Birkhead, Atkin & Møller, 1987*), is supported in flycatchers, in which repeated inseminations from the same individual increased the number of sperm reaching the perivitaline layer (PVL; *Török et al. 2003*). *Savage et al. (2021)* provided evidence that multiple copulations increase the number of sperm reaching the PVL in kākāpō. However, *Birkhead, Atkin & Møller (1987)* concluded that there was no evidence that copulation frequency limited fertilisation across multiple species, and *Hunter et al. (1993)* suggested that the hypothesis could not explain cases where there were high numbers of repeated copulations. Nevertheless, our observed association of higher kākāpō clutch fertility with repeated copulations, together with the results of *Savage et al. (2021)*, suggests that the fertility assurance hypotheses for repeated copulations applies to kākāpō.

The fertility assurance hypothesis is also supported by the result of increased clutch fertility from copulations with different male kākāpō. With no male parental care in kākāpō due to their lek breeding system, there are no clear benefits from increased access to resources from having multiple mates, which is one proposed explanation for polyandry (*Reding, 2014*; *Kempenaers, 2020*). Instead, improved fertility is likely to be a driver for polyandry in kākāpō (*Parker & Birkhead, 2012*). There is support for this from observations of the passerine blue tit (*Cyanistes caeruleus*), in which extra-pair copulations appear to be used to ensure a higher likelihood of fertility when a partner is infertile (*Schmoll & Kleven, 2016*; *Santema, Teltscher & Kempenaers, 2020*). This effect is also likely to apply to lekking species, as copulating with a single male, which might be infertile, has a higher risk of clutch infertility.

### Sperm competition

Competition between sperm from different males in the female reproductive tract might also be important for increasing egg fertility in kākāpō through post-copulatory sperm selection (*Birkhead, Atkin & Møller, 1987*; *Pizzari & Birkhead, 2000*; *Calhim et al., 2008*; *Santema, Teltscher & Kempenaers, 2020*). Evidence supporting this 'sperm competition hypothesis' in kākāpō is provided by sperm morphology. *Carballo et al. (2019)* demonstrated that parrot species which are gregarious, sexually dichromatic and/or have a high level of extra-pair paternity all have longer sperm than monogamic psittacine species, indicating a higher level of sperm competition. Their results therefore support the hypothesis that variation in sperm morphology is driven by sperm competition in psittacines, as it is in passerines. Interestingly, *Carballo et al. (2019)* also demonstrated that kākāpō sperm is longer than many other parrots and is in the range of species with a high level of sperm competition. This suggests that the kākāpō has a naturally high level of sperm competition, which is in accordance with their polyandrous lek breeding system.

Further support for the sperm competition hypothesis driving female kākāpō to copulate with multiple males is provided by the incidence of mixed paternity broods. Under the hypothesis, copulating with multiple males should be common, but mixed paternity within broods should be rare. This is because copulation with multiple males is assumed to be driven by post-copulatory sperm assessment—for example, if the initial mate is unlikely to fertilise the eggs due to infertility or insufficient sperm (*Birkhead, Atkin & Møller, 1987*; *Jennions & Petrie, 2000*; *Rivers & DuVal, 2019*). The frequency of mixed paternity is low in kākāpō: only 2% (one out of 63) of clutches produced by natural copulations with multiple males resulted in mixed paternity.

### Mate guarding

Mate guarding can also explain the instances in which females copulated repeatedly with the same male. With competition for preferred males, female kākāpō may monopolise their preferred mate with repeated courtship and copulations, as hypothesised for other species (*Petrie, 1992*; *Hunter et al., 1993*), including lekking birds (*Petrie et al., 1992*). Females of polyandrous species may do this when there is intense competition for males and a low male:female sex ratio, in order to distract the male from another copulation or to reduce the capability of a male to fertilise another female (*Petrie, 1992*; *Hunter et al., 1993*). Additionally, in populations with high genetic variability among males, females may use repeated copulations to mate guard after copulating with a high-value male, to preserve the genetic advantage of their offspring (*Hunter et al., 1993*). These mate guarding tactics may therefore offer advantages over using aggression to deter other females (*Petrie et al., 1992*). *Petrie et al. (1992)* reported that of feral female peahens which engaged in multiple copulations, approximately half copulated repeatedly with the same male, which is a similar proportion to that found in kākāpō in our study (44%).

Copulations with multiple males can also be explained by mate guarding by female kākāpō, which is common in polygamous species (*Birkhead & Montgomerie, 2020*). In a mating system driven by female choice, it could be expected that since females can assess male quality before copulating, there would be little cause for copulating with multiple

males (*Balmford, 1991*; *Rivers & DuVal, 2019*). However, if mate guarding by females takes place, then copulations with multiple males can result from females having to 'wait' to copulate with their preferred male, and copulating with a non-preferred male first. There is evidence for this in other lekking species, in which females which copulate with non-preferred, subordinate or inexperienced males are more likely to copulate with multiple males (*Petrie et al., 1992*; *Rivers & DuVal, 2019*). This suggests that in such systems the cost of copulation is low compared to the cost of not copulating at all (*Rivers & DuVal, 2019*). In addition to evidence from multiple copulations, there is also observational support for kākāpō females practising mate guarding: at least 13 females have been detected at the display sites of males either the night before and/or after copulation (*Joyce 2009*; KRT, 2021, pers. obs.). Leks are usually outside of females' home ranges (*Joyce, 2009*), so their presence at a male's display site before and particularly after copulation is difficult to explain without invoking repeated copulations and/or mate guarding (*Petrie et al., 1992*).

The correlation of the likelihood of multiple copulations increasing with female:male sex ratio is consistent with the hypothesis that there is mate guarding by female kākāpō. As the threat of competition for mates grows with a changing sex ratio, there may be more mate guarding by females through monopolisation of preferred males with repeated copulations and subsequently more instances of females copulating with different males when their preferred choice is not available (*Petrie et al., 1992*; *Rivers & DuVal, 2019*). Similar variations in mate guarding behaviour with changing levels of competition from varying sex ratio are evident in other species (*Grant & Grant, 2019*; *Birkhead & Montgomerie, 2020*).

### Conservation implications of multiple copulation effects

From their sperm morphology, mating system and our finding of lower fertility from single copulations, we speculate that it is usual for female kākāpō to copulate multiple times and with multiple males. The current situation in which females often copulate once with one male (48% of 217 clutches) may therefore represent an abnormal state.

This situation may be the result of management practices, in which the density of kākāpō on breeding islands (15–20 ha/bird; *Whitehead et al. 2011*) has been limited to reduce the likelihood of male deaths from fighting, to ensure sufficient habitat for females, and to reduce nest interference. If the subsequent density of kākāpō was lower than their natural state, particularly for females, this may have resulted in fewer multiple copulations. Coupled with possible behavioural deficiencies in hand-reared males, this could have led to reduced sperm competition and lower fertility in the contemporary population.

Having sufficient males available at leks was previously assumed to be important to encourage females to visit and mate, but now takes greater significance in ensuring sufficient sperm competition by encouraging repeated copulations and multiple mates. Kākāpō sites should therefore be stocked with high densities of breeding males, while recognising that too many males on leks can lead to higher mortality among males due to fighting. However, the potential impact of female density on fertility, not previously considered in management strategy, appears to be more important than that of male density. Female densities should be kept as high as the habitat can support, with a high

female:male adult sex ratio. There is no evidence of reduction in the number of multiple copulations at high sex ratios, so it appears that adult female:male ratios could be at least as high as 1.6:1. However, this must be balanced against ensuring that nesting females have sufficient quality habitat to enable them to rear chicks in nests.

The optimal sex ratio for kākāpō is unknown, but the only remnant population with both sexes had a male bias of 2:1 (*Powlesland et al., 1995*), which was relieved once the threat of predation was removed and optimised supplementary feeding was introduced (*Clout, Elliott & Robertson 2002*; Data S3). Wild bird populations tend to have male-biased adult sex ratios, but there is evidence that a female bias is normal in lek species such as capercaillie (*Tetrao urogallus*), great bustard (*Otis tarda*) and hummingbirds (*Donald, 2007*). Some populations of these species are heavily male-biased (female:male sex ratio from 1:1.4 to 1:15), but this may be a result of sampling biases or sex-dependent survival in threatened populations (*Mollet et al., 2015*; *Santorek et al., 2021*; *Jiménez et al., 2022*). Indeed, male sex bias may reflect population vulnerability: it is common in small and fragmented populations (*Dale, 2001*); increases with species' IUCN threat status (*Donald, 2007*); and population models and viability analyses for lekking species shows that extinction risk is lowest with a female sex bias (*BessaGomes, Legendre & Clobert, 2004*; *Morales, Bretagnolle & Arroyo, 2005*). So examples of sex ratio from other birds, including lek species, further support the need to maintain a female sex bias in kākāpō.

Artificial insemination should also be continued in kākāpō, as a way to introduce sperm competition when females copulate with only one male. Increasing sperm competition may be as important as the primary reason artificial insemination was initiated in kākāpō, which was to override any natural copulations with a genetically unsuitable (*i.e.,* closely related) mate.

## Age effects

There was no strong impact of the age of either the clutch mother or clutch father on clutch fertility, with both variables contributing negligibly to the model variance. Our analyses were limited in their ability to investigate age effects, given the relatively young age of the contemporary population (mean age = 20.9 and 22.7 respectively for females and males in the model dataset). Our conservative estimate that kākāpō of unknown age were 10 years old on discovery may have exacerbated this by underestimating their true age, but we consider this preferable to removing these individuals from the model, which would impact the ability to investigate other variables. It was also not possible to assess differences in fertility between hand-reared and wild-reared kākāpō with increasing age, since all hand-reared kākāpō were under 25 years old. This should be a focus of future analysis when the data set is sufficiently large, since the developmental environment, including rearing method, has been shown to affect reproductive senescence in other bird species (*Balbontín & Møller, 2015*; *Murgatroyd et al., 2018*; *Cooper & Kruuk, 2018*).

Despite the limitations, our finding of no impact of age on clutch fertility is unsurprising considering that factors such as individual condition, food availability and population density can outweigh age effects (*Hammers et al., 2012*; *Oro et al., 2014*). Similarly, that were were no strong differences in the contribution of mother and father age to clutch

fertility can also be explained by kākāpō ecology. Sex differences in senescence are often more pronounced in polygamous vertebrate species, with males tending to have declining reproductive success at an earlier age than females (*Clutton-Brock & Isvaran, 2007*). This is thought to be a result of males being less likely to win fights as they age, and therefore having reduced access to females (*Clutton-Brock & Isvaran, 2007*). This might be expected in kākāpō, with older, less fit males less able to defend their position in the lek and attract females. However, with the 'exploded' lek system in kākāpō (*Merton, Morris & Atkinson, 1984*), direct competition among males may be less important.

## Copulation experience

Copulation experience (in terms of the number of previous copulations observed since recording began) had no impact on clutch fertility in kākāpō for either sex, unlike in other species (*DuVal, 2012*; *Kokko, 1997*). Our data were limited since some individuals will have had copulations before records began, and some later copulation events were likely to have been missed. However, with the advent of automatic mating detection systems these missed copulations will have been few, and the copulation history information of kākāpō is very detailed compared to most other wild bird species.

Our results are in accordance with female kākāpō not preferentially copulating with the most experienced males (Data S1). Some males have displayed for decades, but have never or rarely mated and produced offspring, despite being visited at the lek by females (*Eason et al., 2006*). Conversely, some young males have produced offspring from first-time matings.

## Inbreeding

Our model showed no discernible effect of parental kinship on apparent fertility, with a very small contribution to the model variance (0.3% of the total ELPD difference). The use of apparent fertility, which combines both 'true' infertility and very early embryo deaths, impeded the ability of our model to determine parental kinship effects. *Savage et al. (2021)* suggest that our sample was likely to be dominated by very early embryo death, which has been attributed to maternal and environmental effects as well as genetic incompatibility (*Savage et al., 2021*; *Assersohn et al., 2021*)–one measure of which is parental kinship.

For the majority of bird species, small sample sizes combined with low rates of infertility have led to reduced statistical power to detect genetic effects on fertility (*Garamszegi, 2016*; *Assersohn et al., 2021*). Our analyses were less impacted by these issues, but were unavoidably restricted by low kinship values and range (0.0–0.265; median = 0.0074; Fig. S2). This was perhaps at least partially a result of genetic management methods such as translocations reducing the likelihood of closely related matings.

However, a study of whooping cranes showed lower parental kinship values and a lower spread (range = 0–0.125; median = 0.0), yet still detected a strong association between parental kinship and apparent fertility (*Brown, Keefer & Songsasen, 2019*). It is unclear why this was not the case with kākāpō, although their different breeding ecology could have led to a different relative contribution of genetic and behavioural effects.

*Jamieson & Ryan (2000)* also reported that higher apparent infertility of takahē on islands compared to their mainland counterparts was at least partially attributable to

genetic factors. However, environmental factors were considered to dominate in takahē fertility, and both the whooping crane and takahē studies did not distinguish true infertility from early embryo death (*Assersohn et al., 2021*).

The results of most other studies assessing effects of parental kinship on fertility cannot be compared to ours, since they use different measures of reproductive success, such as fledging rates (*Morrison, 2020*). However, our results still suggest that parental kinship is not a strong driver of early reproductive failure in kākāpō, relative to the behavioural effects.

Future studies should more closely examine the relationships between other measures of genetic incompatibilities and low rates of fertility in kākāpō. For example, very early embryo death can also be attributed to gross chromosomal abnormalities (*Assersohn, Brekke & Hemmings, 2021*) which would not have been detected in our study.

### *Sperm quality*

Many male kākāpō in the contemporary population have poor sperm quality, with low concentration and a high frequency of morphological abnormalities (*White et al., 2014*). This is quite unusual for polyandrous parrots. *Bublat et al. (2017)* demonstrated that *Eclectus* parrots, which also have a polyandrous breeding strategy, had a high sperm density, very high total sperm count and few morphological issues compared to monogamous macaws, which had a low sperm density, low total sperm count, lower motility and many altered sperm cells. The authors speculated that sperm competition in polyandrous birds is an evolutionary force for high semen quality. *Calhim, Immler & Birkhead (2007)* also suggested that sperm competition can lead to convergence to an optimum sperm morphology within a species. Therefore the low semen quality and quantity found in the contemporary kākāpō population is not expected from their breeding biology, and may instead be due to other reasons such as inbreeding (*White et al., 2014*) or diet.

Recent evidence suggests that male sperm quality may not be such a limiting factor in kākāpō fertility. The microscopic egg analysis of *Savage et al. (2021)* showed that the true egg infertility rate in 2019 was 14%, rather than the 52% assumed. Infertility was still higher in males than females (17% and 2% respectively), but this suggests that embryo deaths, rather than insufficient sperm reaching the egg, are the biggest factor in kākāpō infertility. It is however still possible that sperm abnormalities could be a result of genetic defects which in turn cause embryo deaths.

## Environmental effects

The year random effect in the fertility model accounted for only a relatively small amount of the total variance compared to the random effects of clutch father and mother identity (8% of total ELPD variation for year; 50% and 12% for father and mother respectively). This suggests that variation among years was less important than among individuals (particularly the clutch father), and that unmodelled individual effects dominated unmodelled inter-annual ones. Factors which varied among years would have included environmental factors such as climatic conditions, which may affect fertility, although this is poorly studied in wild species (*Walsh et al., 2019*). Inter-annual variation would also have occurred in food

supply, particularly rimu abundance and whether ripe rimu fruit was available. Rimu abundance is correlated with clutch size in kākāpō (*Harper et al., 2006*), but our results indicate that it is not strongly associated with clutch fertility, nor are other environmental, dietary or climatic variations.

### Other species

The implications from this study, particularly the impacts of hand-rearing, can also be considered for conservation programmes of other species. In a review of global psittacine re-establishment projects, *Joustra (2018)* reported that nearly a quarter (24%) used hand-reared individuals, with two-thirds of those relying on them entirely. Although there are widely-reported negative impacts on behaviours such as reduced predator avoidance, increased human interactions and aggression toward or avoidance of conspecifics (*Carrete & Tella, 2015*; *Utt et al., 2008*; *Joustra, 2018*), further attention should be paid to the more subtle but potentially more damaging impacts on fertility.

## CONCLUSIONS

Our study suggests that some aspects of conservation management have inadvertently affected kākāpō productivity by reducing clutch fertility. The management intervention of hand-rearing, while undoubtedly increasing chick survival, has decreased clutch fertility. The sex difference in this effect indicates that hand-rearing affects copulation behaviour in males more than females, in accordance with imprinting behaviours found in hand-reared male but not female kākāpō. The evidence that female copulation behaviour affects clutch fertility and is in turn affected by adult sex ratios, together with sperm morphology and a mating system which indicates high levels of sperm competition, suggests that current kākāpō copulation frequencies are lower than those previously selected for. This effect is perhaps a result of low population size and may have been compounded by management of population densities.

That female copulation behaviour affects fertility in the lek-breeding kākāpō also has implications for hypotheses for polyandry and repeated copulations. Our results, combined with those on kākāpō sperm morphology, indicate that this behaviour is driven by high levels of sperm competition in kākāpō to improve the likelihood of fertilisation. The increase in multiple copulations with increasing female:male adult sex ratio also provides evidence that female mate guarding occurs in this species.

These combined findings have immediate applications in kākāpō conservation management. Hand-rearing should be limited as much as possible for males; a reversal from previous strategies in which retaining female chicks in nests was prioritised. Population densities should be maximised so that there are sufficient males at leks to ensure adequate mate choice for females, but such that the female:male sex ratio is kept as high as the habitat can support. Artificial insemination should also be continued, to ensure sufficient sperm competition and increase founder representation.

As a growing kākāpō population provides a larger breeding data set, these analyses should be extended to further investigate impacts on fertility. It is particularly important to assess whether the timing of hand-rearing influences fertility. The effects of age, and its

interaction with hand-rearing, should also be re-assessed when there is a wider age range. With a rich genomics data set available for kākāpō (*Guhlin et al., 2022*), the relationship between fertility and measures of genetic incompatibility beyond parental kinship should also be explored. Finally, the findings of this study indicate the critical importance of collecting detailed longitudinal data, and investigating similar impacts of hand-rearing and sex ratios in other threatened bird species.

## ACKNOWLEDGEMENTS

Conservation management of kākāpō is led by the Kākāpō Recovery Programme of the New Zealand Department of Conservation (NZDOC), in close partnership with Ngāi Tahu, the largest Māori iwi (tribe) of the South Island of Aotearoa/New Zealand. The intent is to restore the *mauri* (life force) of the species by returning them to their original range on mainland Aotearoa. This study relied upon the observational data collected and managed over more than 30 years by the kākāpō Recovery Team. Members of this team who collected field data from 1990–2019 are listed below. Thanks are also due to the many volunteers, veterinarians and others who contributed to kākāpō conservation over this period. Huge respect and admiration are especially deserved by the personnel who worked under difficult field conditions without the benefit of remote monitoring methods during periods of low kākāpō productivity from the 1970s to early 2000s. Particular thanks also go to Daryl Eason, Graeme Elliott and Ron Moorhouse for the generation, maintenance and accessibility of this data set. Lydia Uddstrom provided helpful comments on the manuscript. The genetic data relied upon the Kākāpō125+ Project, which generated genomic sequences for all living and recently-deceased kākāpō. The generation and availability of these data owed much to the Genetic Rescue Foundation, Science Exchange and Experiment.com who coordinated and provided funding; Ngāi Tahu who provided advice on Mātauranga Māori (indigenous knowledge) and cultural safety and provided governance; Bruce and Fiona Robertson (University of Otago, Dunedin, New Zealand) who performed DNA extractions; Erich Jarvis (Rockefeller Institute, NY, USA) and Jason Howard (Duke University, USA) who provided genetic advice; many staff members at NZDOC who collected samples; and Genomics Aotearoa who provided advice on conservation genetics and governance. The authors wish to thank two anonymous referees and editor Donald Kramer for the suggestions which greatly improved this manuscript.

Kākāpō Recovery Team field staff Jan 1990–May 2019:

Gary Aburn, Jacinda Amey, Mike Anderson, Karen Andrew, Lisa Argilla, Brent Barrett, Jack Bauer, Chris Birmingham, James Bohan, Liam Bolitho, Dana Boyte, Nichy Brown, Rhys Buckingham, Jo Carpenter, Matt Charteris, Gideon Climo, Stu Cockburn, Ros Cole, Ruth Cole, Rose Collen, Jodie Crane, Dave Crouchley, Andrew Digby, Phred Dobbins, Chris Dyson, Daryl Eason, Graeme Elliott, Melissa Farrimond, Margie Grant, Glen Greaves, Brett Halkett, Jeff Hall, Jason Hamill, Kaitlyn Hamilton, Rose Hanley-Nickolls, Grant Harper, Sam Haultain, Bryony Hitchcock, Rebecca Hohnhold, Stephen Horn, Dean Jakings, Paul Jansen, Bronnie Jeynes, Paul Johnston, Jo Joice, Nik Joice, Leigh Joyce, Sarah Kivi, Sara Larcombe, Jo Ledington, Malcolm Lightband, Matt Low, Karin Ludwig,

Jinty MacTavish, Jason Malham, Anton Marsden, Phil Marsh, Anja McDonald, Kate McInnes, Kate McKenzie, Letitia McRitchie, Nick Mein, Don Merton, Clare Miller, Ricki Ann Mitchell, Freya Moore, Ron Moorhouse, Alan Munn, Lyndsay Murray, Julie Newell, Errol Nye, Jake Osborne, Em Oyston, Nadine Parker, Vivienne Parker, Joanne Paul-Murphy, Brodie Philp, Ursula Poole, Ralph Powlesland, Tony Preston, Tim Raemaekers, Tristan Rawlence, Clio Reid, Hayley Ricardo, Jenny Rickett, Andy Roberts, Sarah Roberts, David Rodda, Rachel Rouse, Malcolm Rutherford, Rachael Sagar, Alyssa Salton, Anne Schlesselmann, Alisha Sherriff, Elton Smith, Vanessa Smith, Nigel Stevenson, Caitlyn Thomas, Theo Thompson, Nick Torr, Jason van de Wetering, Maddie van de Wetering, Nicki van Zyl, Deidre Vercoe, Jen Waite, Jess Wallace, Richard Walle, Jim Watts, Kerry Weston, Jo Whitehead, Amy Whitehead, Liz Whitwell, Bonnie Wilkins, Rebecca Wilson, Jo Wright, Rebecca Wu

### Funding

Andrew Digby, Daryl Eason, Deidre Vercoe, Michael Lierz, Tineke Joustra and Caroline Lees were supported by the Kākāpō Recovery Programme, which is funded by the New Zealand government, public donations and commercial partners. Michael Lierz was also supported by the European Association of Avian Veterinarians. Alejandro Catalina was funded by the Finnish Center for Artificial Intelligence (FCAI) and supported by Aalto-Science IT project. Stephanie Galla and Tammy E. Steeves were funded by the Ministry of Business, Innovation and Employment (MBIE) Endeavour Fund (UOCX1602, awarded to Tammy E. Steeves). Stephanie Galla was also supported by a National Science Foundation Track 2 EPSCoR Program under award number OIA-1826801. Lara Urban was funded by a Feodor Lynen Research Fellowship provided by the Alexander von Humboldt Foundation. Marissa F. Le Lec was also supported by a University of Otago doctoral scholarship. Joseph Guhlin, Marissa F. Le Lec, Lara Urban, Tammy E. Steeves and Peter K. Dearden were supported by Genomics Aotearoa through their High-Quality Genomes and Population Genomics project. Tane Davis was supported by Te Rūnanga o Ngāi Tahu. There was no additional external funding received for this study. The funders had no role in study design, data collection and analysis, decision to publish, or preparation of the manuscript.

### Grant Disclosures

The following grant information was disclosed by the authors:
The New Zealand government, public donations and commercial partners.
The European Association of Avian Veterinarians.
The Finnish Center for Artificial Intelligence (FCAI).
The New Zealand Ministry of Business, Innovation and Employment (MBIE) Endeavour Fund: UOCX1602.
National Science Foundation Track 2 EPSCoR Program: OIA-1826801.
Alexander von Humboldt Foundation.
University of Otago doctoral scholarship.

Genomics Aotearoa High-Quality Genomes and Population Genomics project.

## Competing Interests

Lara Urban, Marissa F. Le Lec, Joseph Guhlin, Peter K. Dearden and Tammy E. Steeves were financially supported by Genomics Aotearoa.

## Author Contributions

- Andrew Digby conceived and designed the experiments, performed the experiments, analyzed the data, prepared figures and/or tables, authored or reviewed drafts of the article, and approved the final draft.
- Daryl Eason conceived and designed the experiments, performed the experiments, analyzed the data, authored or reviewed drafts of the article, and approved the final draft.
- Alejandro Catalina analyzed the data, authored or reviewed drafts of the article, and approved the final draft.
- Michael Lierz conceived and designed the experiments, authored or reviewed drafts of the article, and approved the final draft.
- Stephanie Galla analyzed the data, authored or reviewed drafts of the article, and approved the final draft.
- Lara Urban analyzed the data, authored or reviewed drafts of the article, and approved the final draft.
- Marissa F. Le Lec analyzed the data, authored or reviewed drafts of the article, and approved the final draft.
- Joseph Guhlin analyzed the data, authored or reviewed drafts of the article, and approved the final draft.
- Tammy E. Steeves analyzed the data, authored or reviewed drafts of the article, and approved the final draft.
- Peter K. Dearden analyzed the data, authored or reviewed drafts of the article, and approved the final draft.
- Tineke Joustra analyzed the data, authored or reviewed drafts of the article, and approved the final draft.
- Caroline Lees analyzed the data, authored or reviewed drafts of the article, and approved the final draft.
- Tane Davis conceived and designed the experiments, performed the experiments, authored or reviewed drafts of the article, provided cultural guidance and governance, and approved the final draft.
- Deidre Vercoe conceived and designed the experiments, performed the experiments, authored or reviewed drafts of the article, provided governance, and approved the final draft.

## Animal Ethics

The following information was supplied relating to ethical approvals (i.e., approving body and any reference numbers):

New Zealand Department of Conservation Animals Ethics Committee. The data used in this study were collected as part of routine kākāpō conservation management conducted by NZDOC as required by the New Zealand Conservation Act (1987), and so this study was exempt from the requirement of animal ethics approval under NZDOC's obligations to the New Zealand Animal Welfare Act (1999).

## DNA Deposition

The following information was supplied regarding the deposition of DNA sequences:

The genetic sequencing data are available by application to a data access committee of the New Zealand Department of Conservation (DOC) and Te Rūnanga o Ngāi Tahu. Applications can be submitted at the Kākāpō125+ web page (https://www.doc.govt.nz/our-work/kakapo-recovery/what-we-do/research-for-the-future/kakapo125-gene-sequencing/request-kakapo125-data/) or at the Genomics Aotearoa repository where the data are held (https://repo.data.nesi.org.nz/TAONGA-KAKAPO). Enquiries should be directed to the Kākāpō Recovery Team at kakapogenomics@doc.govt.nz.

## Data Availability

The raw data are available in the Supplemental Files.

## Supplemental Information

Supplemental information for this article can be found online at http://dx.doi.org/10.7717/peerj.14675#supplemental-information.

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
