# Peer review of "Hidden impacts of conservation management on fertility of the critically endangered kākāpō"

_PeerJ, doi:10.7717/peerj.14675_

## Round 0.1 · original submission · Minor Revisions

I apologize for the delay in completing my review. The reviewers were very prompt, but I was delayed by other responsibilities.

Both reviewers considered this a valuable study, and I also found it interesting and useful. Overall, it is very well written, but too detailed in places and not always clear. Reviewer 1 provided numerous detailed comments, mostly related to clarity of the presentation but also questioning some of the peripheral analyses. When I invited Reviewer 2, I specifically requested that he/she examine the analyses because of his/her experience in this area. While the reviewer did not specifically mention it in the comments, they did indicate in their notes to me that they had spent considerable time examining the Bayesian methods and considered them to be appropriate and well interpreted. Reviewer 1, while indicating that he/she was not an expert in this type of analysis, thought the decisions made appeared to be appropriate. Although the suggestions are numerous, the revision is ‘minor’ because they do not require major reanalyses or reinterpretation.

Please note also that the formatting does not follow PeerJ instructions to authors. They specifically request that you not right justify and do not embed tables and figures in the text. I also find it easier to edit manuscripts with 1.5 or double spacing.

Below, I have provided my own comments. You may treat these as a third review, making changes where the comments are valid and explaining why where they are not valid. I have also provided an annotated pdf with some minor grammatical suggestions.

Editor’s Comments
Abstract
L37. This sentence contrasts ‘female’ with ‘father’ rather than female and male or mother and father. This makes it ambiguous if father refers to father of the female or father of her clutch.
L41. ‘Female mate guarding’ could be misinterpreted because it could imply guarding of a female by a male or guarding of a male by a female, and the former is (probably) more common. I suggest ‘mate guarding by females.
L41. Also, infertile clutches don’t have fathers, by definition. Perhaps, specify mate or male.
L46. This is the first mention of sperm morphology and was not a direct contribution of your study. Do you need to mention it here? If it is important, you need to clarify what you are referring to.
L48. ‘High number of clutches from a single mating’ is ambiguous. It could refer to one mating that leads to multiple clutches rather than your intended meaning that many clutches are derived from only a single mating.

Introduction
I agree with Reviewer 1 that the Introduction is too long. The first paragraph is fine. For the second paragraph, I suggest removing the sub-heads and greatly condensing the information on L72-168 into a single paragraph. Focus on what is needed for the reader to know the background and knowledge gap(s) addressed by your study. Next, you could have a paragraph introducing the kakapo, its conservation status and relevant history, its reproductive biology, social system and diet, incorporating relevant information from the separate paragraphs on L169-263 as well as L267-289 and 323. The next paragraph could apply the general background from the second paragraph to potential explanations for low reproductive productivity in the kakapo, more concisely than the present text and ending with a specific statement of objectives of this study.
L130. For most readers, I think a short explanation of how sex ratio affects mate guarding which affects the number of mates is required. Reviewer 2 also makes this point.
L210. Elaborate slightly to allow reader to understand how female choice at leks influences the ‘necessity’ of polyandry.

Materials and Methods
As suggested above, remove all the background biology to the paragraph in the Introduction and focus on the methods relevant to this particular study in this section. An alternative would be to have a much briefer introduction to the kakapo in the Introduction and provide the relevant biology and conservation details in a dedicated, first sub-section of Methods.

Results
Fig. 1. The y-axis label seems very small and lower case does not match the caption. Try printing this and the other figures reduced to the size they would appear on a printed page to see if a reader of the printed version could easily read the axis labels and numbers.
Fig. 2. Is sd, standard deviation? If so, isn’t it usually capitalized. Define ROPE.
Fig. 3. I am not familiar with p.s. and p.d. Please check whether these are standard abbreviations that normally do not require defining. Define ROPE.
Fig. 5. I think that this figure would be clearer with three panels rather than two, a box around each panel and tick marks on the axes. Also, the fonts for axis labels and numbers seem very small.
L539 and elsewhere. Please check d.o.f. I think degrees of freedom is usually abbreviated d.f. or df in Biology.

Discussion
L623. Reword: morphology doesn’t have a length. Sperm are longer.
L637. Again, readers may not expect mate guarding by females, so you need to be more specific if the details were not clarified in the Introduction.

References
Please double-check your references for consistency in formatting and complete information. I noticed some cases with missing page numbers and use of all capitals in journal article titles. (I did not check all references for these errors.)

Reviewer 1 ·

Basic reporting

While there is much to like about this paper—the figures and tables are excellent, for example—the text needs some work that can be readily addressed on revision. First, I find that te Introductory material reads as if it was written by committee, jumping from topic to topic with no clear thread to lead the reader through the material. I would prefer to see only 3-4 paragraphs clearly outlining a bit of the background for this work and the purpose of the study. The rest could be relegated to a section of the Methods on the natural history etc of the Kakapo or simply omitted. Second, too often you use in-house jargon and noun clusters (e.g., mother rearing origin) that make it very difficult for the reader to figure out what you are talking about. Most of the time I could figure it out but a reader who does not know this subject well will struggle, and the flow of the text is interrupted. While I could figure out what you meant most of the time, your use of the term ‘mating’ is particularly confusing as I think that most of the time you mean ‘copulation. I have identified several instances of this sort of confusion in my detailed comments. Finally, the antecedents of your use of it, this and that is often not immediately clear. Again, I think i was able to figure out what you were referring to most of the time but pausing to do so impeded the the flow of the text.

Experimental design

Considering the rarity of this bird, the dataset and the approach to analysis here is reasonable. The authors are careful to acknowledge the limitations of their conclusions. I am not an expert in the Bayesian analyses employed in this study, but the authors have made the sorts of decisions that I would have made. They have done a good job of explaining their methods and results, and in providing their raw data and R code, though I could not get the code to run without error.

Validity of the findings

One aspect of the findings is convincing and well-supported by the data and analyses—of the variables measured, hand-rearing and multiple copulations (or matings? or partners?) have the largest effect on ‘fertility’ in the kakapo. But the ms is missing two things that I look for in a study—estimates of the magnitude of the effects and a measure of the uncertainty in those point estimates. Thus without effects and uncertainties I cannot evaluate the usefulness of the results, nor will they be very useful to managers who might want to use this information to improve kakapo conservation efforts. In fairness I can figure out relative effects and uncertainties from the graphs, but I cannot see any direct estimates of the effect of, say, how much of the variation in fertility is explained by variation in the number of different males with which a female copulates. I would also like to know how much of the variation in fertility remains unexplained b the variables examined.

DETAILS
29 It? antecedent not clear
35 do you mean ‘previous’ mates and mating attempts?
36 and 37 not clear what you mean by ‘multiple mating behaviour’
37-43 it seems to me that you are using the term ‘mating’ here to mean copulations. This needs to be more clearly spelled out
42 not clear what you mean by ‘mother rearing origin’
46 not clear what you mean by ‘together with sperm morphology and evidence for mate guarding,’. Diod you find evidence that sperm morphology and mate guarding influence fertility as this is not mentioned in your summary of Results.
53-55 this statement is too vague to be useful here
74-75 this is not particularly clear. For example ‘raised in captivity by humans’ would seem to include animals in captivity by humans that are fed and protected by their parents. I do not think that that is what you mean but rather ‘fed and cared for by humans rather than the parents of the offspring’
82 not clear what you mean by ‘increased advocacy opportunities’ Give an example or two
83 what is ‘genetic stability’ and why is it important?
87 by ‘differences’ I assume you mean ‘impairment’ as differences could also be positive
101 ‘behavioural differences’ is too vague
105 not clear what you mean by ’strongly significant’. Whether a P-value is low or not is not relevant to the magnitude of the effect. I suggest you refer to the effect size here and not the level of statistical significance. The important point is whether the effect is large or not, significance just depends on the sample sizes and the variances.
128 density is not important. but male-biased sex ratio is. I think you are referring to the number of males relative to the number of females and not to their density per se.
136 this is not true. There is very little clear evidence for reproductive senescence in birds
137 ‘levelling off with age’ is not senescence and I would argue that the evidence for reproductive senescence in birds is rare
140-141 true, but it would be more correct to say ‘most’ rather than many. In fact the the effect of ageing on reproductive output has been measured well in <<1% of extant bird species
143 if this were true you would not have evidence for any reduction in reproductive output
147-148 this is so vague that I am not sire what you mean here
157-168 it would be useful/interesting to have more details here. What happened tro the other 27 individuals? Did they simply not produce any offspring so were not ‘founders’?
234-235 this seems a bit contradictory. If it has not been quantified then there is no evidence at all.
237 does it really optimize productivity? How do you know? Would it be clearer to say ‘in an attempt to maximize productivity’?
285 what does ‘track and bowl’ mean. without some explanation I see no point in mentioning this term
293 not clear what you mean by ‘mating sign’, not how you can infer nesting from ‘daily triangulation’
310 artificial insemination is routinely used in raptor breeding programs and zoos on wild species. I am wondering if you mean this is a first for a free-living bird.
327 I doubt that you tried to maximize the number of chicks per nest and that would also not maximize their growth rates
339 imprinting on humans??
343 at line 330 you say they are not independent until 219 days old so does that mean they were released into the wild before independence. f so how were they cared for after release?
352 no need to say ‘collated in a separate spreadsheet’
354 what do you mean by ‘potential paternity’?
Table 1 caption: explain how you distinguish between apparent and true fertility
370-371 this seems both arbitrary and problematic to me. Thus if a bird was hand-reared for 10 days it was not considered to be hand-reared whereas if it wass hand-reared for 11 days it was considered to be hand-reared? why not include the number of days of hand-rearing in your models or show that there is a clear difference between birds hand-reared or not in the number of days of hand rearing. You could also set a half dozen different criteria for your classification to se if that has any effect on the results.
383 might it be best to cal these ‘minimum ages’?
386 what justification is there for this assumption that errors would be relatively small?
387 even the full dataset is too small for ‘robust analysis’. I would suggest comparing results with and without those birds with estimated minimum ages
428 not clear what you mean by ‘the number of different males the female mated with to produce the clutch’. Is this the total number of copulations before the clutch was produced, or the number of males that the female copulated with before gg laying, or the number of males that had paternity in the clutch?
443-445 this is not at all clear to me. Was fertility status = 1 only if all eggs were fertile, or if any of the eggs were fertile? Also the Fisher exact test does not really answer the question you are trying to address as I think it tells you only that eggs that are fertile are more likely to occur in clutches with other fertile eggs. It does not tell you whether the fertility of each egg is independent of the fertility of other eggs on the clutch. It seems to me that you can get around this problem by using a logistic model structure where the numer of fertile and infertile eggs per clutch is modelled.
453 average number, not overall
516 looking at Fig 1 it is impossible for me to tell if multiple mating explains (i) most of the variance in the response variable (i.e near 100%, which I doubt), (ii) more of the variance than all of the other fixed effect predictors put together (which I also doubt), or (iii) more of the variance than any other predictor (which seems to me most likely). Please clarify
519 it is highly unlikely that any PPDs are actually zero in any model. More likely that they are very small.
533-534 can you put some numbers on the terms ‘higher probability’ and ‘higher likelihood’. For example higher probability of 75% compared to 70% is potentially much more interesting and useful than 10% compared to 5%
536 by how much?
538-543 with your small sample sizes there is likely to be larger confidence limits on those correlation coefficients so those should be reported as well as testing, for example, whether 0.92 is actually convincingly distinguishable from 0.61 with these data
Fig 2 and 3. can you put units on those ‘Possible parameter values’? Since you standardized are these in SD units
Fig 4 caption needs to define clutch fertility.
549 is this generally true or just among the factors you looked at? Without some, preferably quantitative, assessment as to how much of female infertility is due to these factors, as mentioned above, I cannot assess the validity of this statement.
554 again, how much of an effect is it. Whether it’s statistically significant or not is irrelevant here. We need to know the effect size and the uncertainty around that point estimate. There is an attempt t show this in Fig 4 but the units of ‘fertility’ are not clear. By whatever measure that is, hand-rearing does seem to have a consistently lower fertility than wild-rearing but the uncertainty in those estimates is large enough that some caution is called for.
567 imprinted on humans
Figure 5 not clear what you mean by multiple matings—copulations with more than one male, or multiple paternity in a clutch? So is it the clutches that resulted from multiple copulations resulting in mixed paternity, or the female that engaged in copulations with multiple males?
620 I would suggest a little more caution here “thus indicating a higher level of sperm competition” as other explanations are possible
663-665 alternatively females in better health/condition—and thus more likely to have viable ova—might simply copulate more frequently. You do not seem to have data to test that reasonable alternative.
698-700 this seems an odd suggestion given the importance of hand-rearing for rescuing populations. Your data suggest only that hand-rearing practices need more careful study and modification to reduce adverse effects.
703 ‘model clutch data’??
720 missing verb
722 female
727-757 these effects of kinship presumably depend on the level of kinship between parents, especially in relation to the ‘normal’ level of parental kinship in a viable wild population. I realize that you won’t have such data from a wild population but some assessment of kinship compared to other species is warranted here
759-767 see also Calhim S et al. (2007) Postcopulatory sexual selection is associated with reduced variation in sperm morphology. PLoS ONE 2: e413.
768-772 sperm abnormalities could also indicate DNA defects that would cause embryo mortality after successful fertilization
774-776 some quantitative comparisons would be useful here
Supporting files: The R notebook that I extracted from your Supplemental_Data_S3.html file would not run as it gave the following error at CHAIN 2: Error in unserialize(socklist[[n]]) : error reading from connection. I have run STAN models without a problem but I am by no means an expert in the implementation of bmrs so I could not readily fix that problem. I suggest that you thoroughly test your code on different computers to make sure that it runs efficiently. That code is not particularly easy to follow as it seems to be minimally annotated.

Additional comments

This is potentially a very useful study to inform future conservation efforts on this endangered species and can serve as a model for collecting and analyzing data of this sort in other conservation efforts. As I have mentioned elsewhere in this review, however, I feel that both the clarity of the presentation and the transparency of the results could be improved to make this study as useful as possible. I appreciate the inclusion or data and R code by the authors, but this material is also limited in its usefulness and accessibility. I really like your use of rmdformats to format your code and output in an html file in a very attractive and accessible document. However, that file and the data files should, in my opinion, also be provided in a more open format that will allow the user./reader to open the datasets in any software, and to run the R code directly. I was able to extract your R notebook (Rmd file) from your html file but I would suggest supplying that Rmd file as well. I was also able to extract your data from your rds files and save them in a more open format (csv) and would suggest you do that as well. I have also provided an extensive list of detailed comments that provide specific examples of some of the difficulties that I had in fully understanding your ms.

Reviewer 2 ·

Basic reporting

no comment

Experimental design

no comment

Validity of the findings

no comment

Additional comments

I found the manuscript entitled “Hidden impacts of conservation management on fertility of
the critically endangered kakapo” to be a very relevant and well-written study. My only comments are minor suggestions to improve wording or readability:

L130: This is the first mention of mate guarding outside of the abstract, and the topic returns in the discussion on L631-638. However, I was unclear whether the reference to mate-guarding here referred to females guarding displaying males, or males guarding one (or several) females, preventing them from investigating other displaying males. That question was obviously resolved in the next paragraph (beginning L 639), but I wonder if perhaps it would be helpful to specify that it is guarding by females being discussed, at least at the beginning of the paragraph that begins L631, and perhaps in the Introduction as well. In fact, much of the material in L639-643 might be better in the Introduction, because it also provides a clearer explanation for the motivation behind examining sex ratio in the first place.

L306: Should this read either “by genotyping” or “by sequencing”?

Figure 4: The caption refers to “filled circles” twice, once for the opaque model predictions and once for the semi-transparent observations. This could cause confusion, and it may be better to refer to the observational data as something like “smaller filled circles” or “faded circles” in the caption.

L627-628: It is not clear from the information provided why mixed-paternity broods should be rare. This is probably not a very important point, so perhaps not worth a long digression, but if it is possible to provide a bit more detail on Rivers and DuVal’s results it would help the reader follow the argument.

L722: typo: “females” should read “female”

L729: typo: should read “…to be dominated…”

L764: typo: “cell” should read “cells”

L792: Another benefit of the Bayesian approach taken in this study is that such reanalysis in future years will be very straightforward. This may be worth pointing that out to readers who might be considering whether to adopt a similar approach for their systems.

I congratulate the authors on their analysis and manuscript and look forward to seeing it in print.

---

## Round 0.2 · Minor Revisions

Thank you for your detailed reply to the reviewers and to me. Only Reviewer 1 was available to review the revised manuscript. The most important issue raised by the reviewer is a concern that the conclusions about hand-rearing and population sex ratio are insufficiently robust to influence policy because of the analytic approaches to clutch fertility and age and the use of clutch fertility as a measure of productivity. I was more convinced by your results, but the reviewer has a much deeper understanding of multivariate methods than I do, so it is important that you address these concerns seriously. I also recognize that conservation decisions may need to be made without ‘perfect’ knowledge of the implications, and it seems to me that your evidence indicates a change in protocols, even if it isn’t completely air tight. What I would like to see is a much more self-critical approach to the Discussion, clearly indicating both the strengths and weaknesses of your conclusions. In the Conclusions section or later in the Discussion, you can address policy with the probably implications of continuing with present policy as well as the of changing.

The reviewer is also concerned that there is no clear logical explanation for the patterns seen. I don’t agree with the implication that one needs a clear mechanism in order for the empirical pattern to be valid. Nevertheless, I think that you have some ideas about this that could be presented more explicitly.

Although it is possible that some of the reviewer’s concerns will lead you to change analyses, I believe that the basic structure and conclusions of the manuscript will stand, so I am designating the changes as ‘minor’.

I read the manuscript with an eye to grammar and clarity as well as content. Some of the points raised above are elaborated in my following comments. You may treat these as a third review, making changes when you accept the point and explaining your decision if you think I am wrong. My suggestions regarding grammar and style are indicated on the attached pdf. You need to respond to these in the rebuttal only if you disagree.

Editor’s Specific Suggestions and Comments

Title: The reviewer suggests removing ‘hidden’. I can see why you have included it and why they suggested removing it. I will accept either.

Abstract:
L30. Add a brief description of what kind of organism this is, especially background relevant to understanding the abstract (e.g., parrot, flightless, nocturnal, long-lived, endemic to New Zealand, lek mating system) and give family after scientific name. Probably the sentence will need to be restructured.

Introduction
L90. 'Mating experience' is ambiguous; here, you could state specifically whether you are referring to individuals that have had more mates or more copulations or both, and whether it applies to both males and females. It is also not clear precisely what you mean by ‘productivity’; is this a standard term in reproductive biology/population ecology? The ‘also’ implies that implies that you may consider it to mean the same as ‘reproductive output’, but even that is a bit ambiguous because it is not clear whether the stage at which it is estimated (fertilized eggs to independent young) is implied. Please check the terms to be sure you are using them correctly and consistently.

L112. At the end of this paragraph, you could add a sentence to clarify the ‘mate’ ambiguity. For example, ‘Therefore, we refer to males as mates if they have copulated with a female at least once and do not imply a longer-term pair bond or association.’

Methods
L159. You have improved the distinction between copulation and male partners (mates) substantially, but some ambiguous wording remains. This line has one example. I have highlighted cases that I noticed, but you should carefully check the entire manuscript including table headings, table bodies, figure captions and figures to find and correct any remaining problems. Also, please check to be sure you use exactly the same term for each variable throughout the manuscript (e.g., L416 seems slightly different).

Fig. S1. This figure is not completely clear. Most journals do not publish figures with grid lines crossing the main body of the figure. In this case, it is particularly ambiguous because you have grid lines at half individuals. I suggest removing grid lines and adding solid axes with tic marks, labelling the y-axis as ‘number of individuals’. The x-axis is not clear because neither caption nor axis label makes the bin size clear. I initially read the tall bar at zero as cases with no days of hand-rearing. Comparing the counts to your text, I see that that was wrong. However, it is not easy to figure out which number of days are binned by each bar. Tick marks at intervals of 5, if that is bin size, with bars based on each gap would make it clearer without needing to label each bar.

L255. I was more convinced than the reviewer of the validity of using a binary variable for rearing status, given the strongly bimodal distribution of hand-rearing durations. However, it might be worth noting in the Discussion that your data do not show that hand-rearing up to 10 days is ‘safe’ because there were only 8 (I think) males reared for this period of which 7 (I think) were hand-reared fewer than 5 days. You should also be specific that the number of days hand-reared does not necessarily correspond to the chick age, which I initially assumed. It could be that some short periods of hand-rearing were outside a critical period so might not have the same effect as the same number of days at another stage of development. This is another point which you should note as a qualification of the strength of your conclusions in the Discussion and a topic for further research in the Conclusions.

L282. Make clear what the time scale is of the ‘cumulative’ number of copulations, breeding season, lifetime, since recording started, or something else. Again, imprecision in this measure might be a source of error to be raised in the discussion of the strength of your conclusions.
L395. This is the number on the whole island? Is the island small enough that these individuals could all potentially interact? Is there a strong time correlate of these numbers, and thus a potential confound, due to the increasing population size and/or management manipulation of sex ratio? Another caveat for the Discussion, perhaps?

Results
The reviewer suggested that Fig. 1 and the text (L403-404) do not agree. I assume that this is because the ELPD difference value for ‘mating’ is similar to that of the next seven variables. If the important information is the magnitude of the differences between successive variables, the figure would agree with the text, but if it is the value read from the y-axis it would not. I can’t quite get my head around the changes between successive variables because it is a ‘difference between differences’. Your Results should clarify how readers should interpret the figure and why.

L435ff and Fig. 5. I don’t fully understand these analyses. They are based on 29 years, but there are only 10 data points. Furthermore, I would have expected d.f. for 10 points to be 9, not 8. Furthermore, if the population is growing consistently, there would be a strong correlation between population size and year, which might need to be considered.

Fig. 1 does not require grid lines in the text but needs well defined axes with tick marks. It should include -60 on the y-axis.

Fig. 2 has similar formatting issues to Fig. 1. In addition, I don’t think I understand the units of the x-axis (0.5*SD). Does this mean that if the axis reads 2, the value is 2 x 0.5 or 1 SD? I think the caption needs to be clearer. I also find the second sentence of the caption awkward and unclear. Consider 'The less a posterior distribution intersects the ROPE (region of practical equivalence, denoted by the shaded vertical bar), the stronger the association of that parameter with fertility. (See Statistical Analyses for details.)' Also, the caption refers to parameters for what are, if I understand correctly, called variables in Analysis section.

Fig. 3. Many of the same issues as in Fig. 2 apply here. Much of this caption is devoted to elaborating the result, unlike previous figures. Please move the text results to the body of the text and focus the caption on what the readers need to know to understand the figure. The seventh line also includes an incomplete comparison which should be completed along with the move to the body of the article.

Fig. 4. Many of the same issues apply. However, in this case, I think that the light vertical grid lines are visually useful. This figure would not be very accessible to someone who was red-green color blind or had only a black-and-white photocopy. You can still use color, but add also another cue such as shape or filled vs. open circles.

Fig. 5. Grid lines and axis issues apply as in other figures. Three panels attached with lines around all 4 sides of each and letters inside, as per PeerJ instructions, would work. The y-axis label is rather long and contains a non-standard abbreviation; I think ‘Proportion of clutches’ would be sufficient. The x-axis numbers should bracket the values, i.e., include 40 for females, 10 and 30 for males, 0.6 for ratio. The x-axis legend could be ‘Number of females’ and ‘Number of males’ for the first two panels. The caption should specify what the shaded area indicates and should not include the last sentence describing the pattern; it is already well described in the body of the text. It is awkward and potentially misleading that the scale for number of males is not the same as number of females. Consider using the same scale; otherwise, point out the difference in the caption.

Discussion

In keeping with the reviewer’s concerns, each section of your Discussion should address first the strengths and weakness of your findings before relating it to the literature and implications for management. I have some specific suggestions of topics you might consider for each section, but your knowledge of the system and potentially critical responses should allow you to know what issues should be mentioned and which need not be.

L448. Is productivity the best word here? Can you be more specific, or do you really want to be general, despite the focus of your study on clutch fertility which is only one component of productivity if the term means something like annual or lifetime reproductive success (to what life history stage?) per female?

Rearing environment
• Should you be more explicit in the Discussion that hand rearing durations are not necessarily the related to hatch day, as I had first assumed? Critical periods for imprinting might make hand rearing for the same number of days more critical as some ages than others. There is probably relevant literature on the time and duration of exposure to an imprinting object in the literature for other species.
• Is there an issue associated with combining completely wild reared with hand reared up to 10 d? Is there a need for future research to check whether even short hand rearing duration is a problem? The number of hand-reared males is very small, especially between 5 and 10 days.
• The reviewer says there is ‘no convincing logical arguments as to why hand rearing should affect male fertility’ and later asks if you can propose a mechanism. Although you hint that performance of copulation or possibly mate recognition might be the problem, I agree with the reviewer that a more explicit statement of the suggestion and evidence available (including possibly evidence from other species) and evidence needed would be helpful.
• The idea that hand rearing may be less of a problem if the chicks are in groups is not clearly developed. It is not clear whether there is evidence of a benefit to rearing in groups.
• Is fertility the right word for male ability to produce a fertile clutch when the barrier is likely to be copulation performance rather than sperm production?
• I don’t see how the takahe example supports a hypothesis about kakapo. The kakapo hypothesis seems to be related to copulatory behavior whereas the takahe example seems to be about parental care of chicks.

Multiple copulations and mates
• The manuscript generally considers the order a) one copulation with one male, b) multiple copulations with one male, and c) multiple copulations with more than one male. Although I can see why you would put multiple males first as the stronger effect, it would make sense for you to address multiple copulations before multiple males both to keep the order consistent and because of the conceptual aspect of increasing one variable at a time. I suggest that you do not subdivide this section, and integrate the discussion of fertility assurance and sperm competition after addressing the evidence for each because the concept apply to both.
• L506. I am not convinced that ‘mate choice’ is the best term to describe patterns in the number of copulations and number of male partners. You do not have evidence that these patterns are related to ‘choice’, mating with one male rather than another. Perhaps ‘mating behavior’ would be a more general term that would include these aspects. If you agree, check the rest of the manuscript to see if you have used ‘choice’ elsewhere for this concept.
• I agree that ‘fertility’ is an appropriate term when referring to the clutch produced by the female.
• For both copulations and mates, briefly review the strength of the statistical evidence for the effect.
• Then, consider potential limitations with the evidence for each effect. Some that might be considered relate to the grouping of the data (not surprising, given the limited sample size, but still to be recognized). I don’t expect you do devote a lot of words to ‘apologizing’ for the limitations, but it is still important to recognize when and how they might affect the conclusions or recommendations.
o Increase in fertility with copulations might not be one vs. more than one but have some higher threshold. For future studies, the change in fertility as a function of copulation number would be of interest.
o Temporal distributions of the copulations (when they mate) might be important in relation to female fertile periods and sperm storage.
o The effect of multiple mates includes multiple copulations. It is possible that effect of mates is partially or total a multiple copulation effect. For future studies, the relationship between number of mates and fertility, controlling for number of copulations would be of interest as well as the pattern of increasing fertility with number of mates. Did you ever present the data on number of mates?
o Clutch fertility is a coarse measure. The proportion/number of fertile eggs could have strong effects on reproductive success unless there are very strong density-dependent effects within a nest.
o You do mention potential biases from assigning paternity to clutches for which it is not known or by ignoring such clutches, but a clearer explanation of the direction of effect and the biases is needed.
o You also mention female condition as an explanation for multiple mating. A bit more explanation of the proposed mechanism is needed, and it should be presented as part of the critical discussion of the evidence before literature comparison.
o L571. I don’t see how 6 cases of mating with one male before and after mating with another or sometimes mating with only one male is sufficient to reject the male quality hypothesis. Adaptive behavior is often less than perfect.
• Effect of sex ratio on multiple mating.
o This topic could also be integrated over multiple copulations and multiple male partners with the above proposed restructuring of the mating section.
o If there was aggregation among years, could this affect the results?
o Again, a critical discussion of the strength of the evidence is needed, especially as this relates to policy implications. What are potential correlates of sex ratio that might confound the simple correlations? For example, is there a change in sex ratio over time as the population developed? Have there been other changes in the environment or conservation procedures over this time? Is the sex ratio of the whole island what the birds experience or is it more local in space and time? For example, is there only one lek attending by all the males most of the time?
o Is the range of variation in sex ratio, 0.6-1.6 relevant (I agree that it is)? How does this compare to natural variation in sex ratio of lekking species? What about the lack of data between 0.6 and 1.1?
o A clearer explanation of how sex ratio could affect multiple mates and copulations is needed. Intuitively, as female:male ratio increases, there is a reduction in relative male availability which should lead to greater competition among females and reduce copulations/female and mates/female. So, why is the opposite pattern observed?
 The suggestion that females mate with a less preferred male while awaiting the availability of a more preferred male makes some sense. Does the distribution of number of males mated with correspond to this explanation?
 I do not find the mate-guarding explanation for multiple copulations as clear. I can see the advantage of multiple copulations for the female, but what benefit is there by restricting access by other females to the same male? If the issue is amount of sperm, it would seem that the female would be better off to use aggression rather than copulations to keep her preferred male from mating with other females.
 You need to explain how observations of the female at the lek before or after copulation provides evidence for mate guarding (L540ff).
• Copulation experience. Although you reject this as having an effect on fertility, you should also (briefly) critically consider the evidence. I don’t think you ever indicated over what time scale the variable was defined (season, life time) and how reliable the values are. Could using an all-or-none variable for clutch fertility affect the statistical pattern?
• Inbreeding. Could you reorganize the paragraph to present the critical discussion of the evidence (e.g., low kinship) first and then relate it to the literature?
• Can the sperm quality discussion be incorporated into the sperm competition discussion to remove this section?

Conclusions
According to PeerJ instructions, the conclusions should not repeat the findings as if they were another abstract but should emphasize implications (perhaps conservation implications could go here) as well as future research needs.

Reviewer 1 ·

Basic reporting

The authors have gone a long way to improving the readability of this ms, largely in response to reviewers’ comments. I still think there are too many noun clusters that can be easily fixed without unduly lengthening the manuscript. Despite their claim, ‘mating’ is not well defined and seems to be conflated with copulations in several places. I particularly appreciated the improved clarity in the reporting of statistical results and the inclusion in the models of paternity estimates for infertile clutches when the multiple fathers’ identities were not known.

Experimental design

The inclusion of paternity estimates for infertile clutches with multiple paternity is a big improvement here. I have some concerns about the scoring of unknown paternities, ages, and clutch fertility but these are easily dealt with using new analyses, or at the very least explaining the limitations of the scoring methods used.

Validity of the findings

I have some serious concerns about the conclusions of this study. I know almost nothing about kakapo nor am I a conservation biologist so I come to this study with no particular biases in those regards. I worry, though, that some of the choices made in the analyses may have biased the results and I see no particular attempt to allay those concerns either by supporting analyses or logical argument. For example, I see no convincing logical arguments as to why hand rearing should affect male fertility, nor why sex ratios should influence productivity, and those are two major conclusions of this study that the authors want to use to influence policy. I worry that the choices made in scoring clutch fertility, using fertility as an index of productivity, and using a minimal estimate for the age of unknown birds, for example, might have influenced the results. If I was a policy maker involved in the conservation of this endangered species, I would want to see more convincing evidence before changing the current policies, including better logic underlying the apparent findings of this study and a more thorough exploration of the possibilities of bias in the analyses.

Additional comments

I was reviewer 1 of your original submission, so I will first adress your rebuttals to my suggestions, then provide a thorough review of your revision as you state that you made substantial changes to the analyses etc. Thank you for your thoughtful responses to my comments.

I see only five places where you disagreed somewhat with my suggestions so I will only address those. First, you mention that the magnitude and uncertainties of the effects were shown in Figs 1-4 and Supplement. I was well aware of that and this is a common practice that I find to be less than adequate. Those magnitudes and uncertainties are much more important (to me) than statistical significance and should be highlighted in the main text where you seemed to want to focus on the significance instead. Second, in a few places you dispute my contention that what you say is unclear. I am not sure why you would dispute my impression as you have no access to what I am thinking. Those statements were unclear to me and I expected that they would be to other readers. Third, for line 327, you say that you did try to maximize the number of chicks per nest but it seems that 3 might have been your maximum. What about 10 chicks per nest and if that was not possible, I think you should say why you could not try more than 3, or that you maximized up to three. Fourth, for line 387 you disagree with my contention that these are not robust analyses. But then you say they are actually robust “ given what is possible with a relatively small population”/ I agree that you have these constraints but that qualifier should be added because these analyses are not really robust BECAUSE of the small sample sizes. Finally, for lines 443-445 you disagree with my comments about the Fisher Exact Test. I have no problem with your explanation but I would contend that your scoring method obscures some underlying and potentially interesting variation. It might be worth mentioning that it is certainly possible to consider the numbers of eggs that are fertile or infertile in a binomial model.

Thank you also for your response to my problems with the R code. I have never had a problem with parallel processing on my computer (10 cores). I can appreciate that there might be system-specific problems so it might be useful to include your system details in your README file

The remainder of this review refers to your revised ms


DETAILS
Title: why ‘Hidden’? All new findings were previously unknown so this seems redundant.

42 this seems counterintuitive to me. If the female:male ratio increases I would have expected fewer extrapair copulations and mates not more.

50 again, why not the result of too few males?

74 why ‘precocial’? This would also be true of species with altricial development. Would it not be more to correct to say something like ‘during chick development to maturity (or independence)’?

104 you seem to want to list the Maori names first so why not Pukenui/Anchor Island? It is also my understanding that the Maori name for Little Barrier is Te Hauturu-o-Toi

123 by ‘breeding’ do you mean ‘copulations’? Incubation and chick rearing are part of breeding in birds.

138 in the wild?

156-179 already it is not clear to me what you mean by ‘mating’. Since this is all determined by remote sensing can you determine if a lek visit involves a copulation. If so I would use the term ‘lek visits’ and ‘copulations’ tpo distinguish the two different sorts of records. It would then also be useful to provide information on how you distinguished a lek visit (minimum duration??) and copulations using those recordings.

173-174 a reference to this newer technique would be useful here

190 were they replaced with artificial eggs to keep the female incubating?

192 as noted above, I would say ‘to increase’ rather than ‘to maximize’

193 how could each chick have ‘multiple foster mothers’? Did you move chicks from nest to nest after the first fostering?

201-202 how does grouping the chicks avoid imprinting? This would not be true for waterfowl, for example.

227 ‘copulated’ not ‘mated’

229-238 it sounds like you assigned a single father to each of these clutches. That would, I think, be a mistake. Can you not use these methods to assign multiple fathers based on the frequency of copulations and the data you have on the males that copulated first, middle and last?

Table 1. Not clear what ‘Matings’ refers to here. Surely it’s not possible to get fertile eggs from zero copulations. or 4 clutches from a single copulation. It might also be useful to have a column for the number of females that produced clutches each year, or a statement that each clutch is from a different female within years

255 this cutoff still bothers me as there is surely a big difference between one and 9 days of hand-rearing. I am not so sure that there is a good reason to simplify statistical analysis here as you seem perfectly willing to use complex analyses in other parts of this paper

272-275 this seems an odd statement to me. How can you be sure that using a minimum age of 10 for >25% of the birds, is unlikely to introduce bias with a species that might breed up to age 40 or more. And as I pointed out above, even with those guesses included, the dataset is still too small for robust analysis. It would be clearer and more correct to say that excluding those birds would result in less reliable conclusions due to the small sample size. I do not see any obvious solution to these issues. One possibility would be to assign an age to those unknown-age birds based on the frequency distribution of know-age breeders, randomly drawing the age of each unknown from that distribution and then repeating that procedure 1000 times to assess the reliability of the results.

281 I cannot tell if mean the copulation history within each year or across all years. It is not clear to me how the paternity analysis influences copulation history. I am guessing that you used paternity testing to indicate that a copulation had occurred even when no copulation was recorded but that should be stated

318 father’s

321-322 ‘included’?? Did it include anything else? And it sounds as if you only considered there to be multiple matings if the female ‘copulated with different males more than once’. Does this mean you did not count them as multipole matings if she copulated with different males only once each?

322 that data for this Fisher Test should be presented here. And, as I have argued above, I think the binary classification might introduce a bias.

336 I am not sure that the imbalance iss an important consideration here and I feel that island should be in the model

337 (and 166) why the Maori name for one island and not the other

387 version 4.1.2 was not released until Nov 2021 so the ref date should be 2021 here

393-394 ‘multiple matings’ should be called ‘multiple copulations’ sad in some cases only a single mate was involved

403 clutch father’s

403-404, and 407 multiple copulations variable

411 I do not see the point of saying ‘significantly’ here

416 which is it, mates or copulations

Figure 1 This plot suggests to me that only father HR has an effect on the model selection that is appreciably different from the effects of all other predictors between ‘Mating (whatever that is) and Mother/Father kinship. This seems at odds with what you say in the text

435-446 I think I see now how this correlation could be positive (see above for my doubts). When female:male sex ratio is high females might guard a male more intensely, resulting in multiple copulations with that male. Is it possible to see if that was indeed the case, or did the number of copulations with all males increase

451 presumably within a given breeding season

Figure 4 I think you mean copulation, not mating. Since you are concerned with productivity in this paper I am not so certain that the differences in actual productivity—number of chicks fledged—would be so clear. For example, hand-reared males might have lower fertility but higher offspring survival. I would like to see you address that.

455 you do not know that it affects productivity only fertility as you narrowly define it

455-491 can you suggest a mechanism by which hand-rearing might influence male fertility. Might it be possible that hand0rearing affects the ability of males to copulate successfully such that a copulation results in little or no sperm entering the female?

506 you have not shown any evidence of mate choice. The number of mates and copulations could simply be a function of lek size etc

512 there is no evidence that it is a KEY driver of polyandry. And Birkhead et al 1987 made no such claim. In fact they rejected that hypothesis with the available data. There is, however, some more recent evidence to support that hypothesis in a few species.

538-540 this assumes, of course, that there is some reason to mate with a non-preferred male at all. Thie logic in this paragraph needs to be better developed

543-544 this is a bit circular to say the least as you have just argued that mate guarding promotes multiple paternity

674 a high incidence of morphological anomalies?

704 I do not see the need for the word ‘robust’ here

Supplements
The supplementary figures and tabel see fine to me but I would recommend compiling them all in a single document with the captions

I had a quick look at the data and code. Thank you for making the data files into csv format. Your paper says “Formal analysis (clutch fertility and demography): AC, AD, DE.” but only AC and AD are listed as authors of the R script. The R code and output also seem OK and ran just fine this time. I would hope that you would put that script, the data files and a comprehensive README file into a public data repository like OSF or Dataverse where they will be curated and available independent of the published paper.

---

## Round 0.3 · accepted · Accept

Thank you for your detailed and thorough responses to suggestions. I have read the rebuttal and the track changes documents and looked at the supplementary figures and tables and consider the manuscript now suitable for publication now.

I found a couple of minor errors in the text which could be corrected during manuscript processing:
L492 ‘any taxon’ not ‘any taxa’
L725 ‘examples . . . support’, not ‘examples . . . supports’

In addition, it might be helpful to add units to the caption for Fig. S2 or the figure itself [age (years), number of copulations, kinship (r)]

I will pass on your suggestions to PeerJ by email, copied to the corresponding author.